# Double helical conformation and extreme rigidity in a rodlike polyelectrolyte

Ying Wang[1], Yadong He[2], Zhou Yu[2], Jianwei Gao[3], Stephanie ten Brinck[4], Carla Slebodnick [1], Gregory B. Fahs[1], Curt J. Zanelotti[1], Maruti Hegde[3,5], Robert B. Moore[1], Bernd Ensing[4], Theo J. Dingemans[3,5], Rui Qiao [2] & Louis A. Madsen [1]

The ubiquitous biomacromolecule DNA has an axial rigidity persistence length of ~50 nm, driven by its elegant double helical structure. While double and multiple helix structures appear widely in nature, only rarely are these found in synthetic non-chiral macromolecules. Here we report a double helical conformation in the densely charged aromatic polyamide poly (2,2′-disulfonyl-4,4′-benzidine terephthalamide) or PBDT. This double helix macromolecule represents one of the most rigid simple molecular structures known, exhibiting an extremely high axial persistence length (~1 micrometer). We present X-ray diffraction, NMR spectroscopy, and molecular dynamics (MD) simulations that reveal and confirm the double helical conformation. The discovery of this extreme rigidity in combination with high charge density gives insight into the self-assembly of molecular ionic composites with high mechanical modulus (~ 1 GPa) yet with liquid-like ion motions inside, and provides fodder for formation of other 1D-reinforced composites.

[1] Department of Chemistry and Macromolecules Innovation Institute, Virginia Tech, Blacksburg, VA 24061, USA. [2] Department of Mechanical Engineering, Virginia Tech, Blacksburg, VA 24061, USA. [3] Department of Aerospace Engineering, Delft University of Technology, Kluyverweg 1, 2629 HS Delft, The Netherlands. [4] Van't Hoff Institute for Molecular Sciences, University of Amsterdam, Science Park 904, 1098 XH Amsterdam, The Netherlands. [5] Department of Applied Physical Sciences, University of North Carolina at Chapel Hill, 121 South Road, Chapel Hill, NC 27599-3050, USA. Correspondence and requests for materials should be addressed to R.Q. (email: ruiqiao@vt.edu) or to L.A.M. (email: lmadsen@vt.edu)

D NA molecules, which act as a storage and transfer platform for the genetic information of life, exhibit a double-stranded helical conformation[1,2]. Further applications of helical macromolecules involve not only enantioselective and asymmetric catalysts for chemical reactions, but also scaffolds and templates for supramolecular self-assembly[3,4]. Inspired by natural multi-stranded helical macromolecules such as polypeptides, collagen, and polysaccharides, researchers are engaged in constructing synthetic polymers with a variety of helical conformations[3–8]. Broadly speaking, the most widely used chemical and physical foundations to build double-stranded macromolecules rely on metal-directed (ligand-containing) self-assembly[7,9], aromatic stacking[10,11], and hydrogen-bonding-driven self-assembly (inter-strand hydrogen bonding)[8,12,13].

The most commonly encountered structural motifs in oligomers that enable double helical formation are based on peptides[14,15], peptide nucleic acids[16], amidinium-carboxylate salt bridges[6,8], and coordination polymers[7,9,17–19]. While synthetic single helical strands are relatively common, despite advances in synthetic strategies, the design of synthetic double helical strands in solution remains a challenge[6,8,20–22]. Isotactic poly(methyl methacrylate) (it-PMMA)[23] forms double helices, but only in the solid crystalline state. In aromatic polymers, the amide linkage functionality is preferred as it participates in H-bonding. Generally, aromatic cores are meta- or ortho-linked to enable a twist necessary to form a helix, also known as a helical foldamer[24–26]. Additionally, the presence of polar side groups such as hydroxy (H-bonding) or pendant aliphatic moieties (steric hindrance) can drive helix formation[27]. To the best of our knowledge, with the exceptions of some oligomers, e.g., Yashima's oligoresorcinol[28] and Huc's heteromeric oligoamides[29], no synthetic motifs have been demonstrated to form double helices in water.

Herein, we describe an aromatic sulfonated all-para linked polyamide that possesses the double helical structure in water, thus leading to extreme axial rigidity of this polymer chain. As far as we know, this is the only instance of a synthetic polymer that can form a double helix in aqueous solution. This all-para, all-aromatic-backbone macromolecule is obtained using a simple, single-step interfacial polycondensation reaction. We verify formation of this double helix via complementary investigations utilizing X-ray diffraction (XRD), [23]Na NMR spectroscopy, and molecular dynamics simulations. This synthetic sulfonated aramid polyanion, poly-2,2'-disulfonyl-4,4'-benzidine terephthalamide (PBDT), can be used to form a unique series of hydrogels[30,31] and ion gels[32,33], which have displayed great potential as next-generation functional materials for batteries, fuel cells, and optical sensors. The high rigidity imparted by the double helix drives important properties in these materials. PBDT is a water-miscible polymer that forms a highly anisotropic lyotropic nematic liquid crystal (LC) phase[34] at concentrations down to exceedingly low values ($C_{PBDT} \geq 0.3$ wt%). Using [2]H NMR spectroscopy and small-angle X-ray scattering (SAXS), we previously investigated nematic LC ordering and polymer chain–chain distance in PBDT aqueous solutions[34]. However, our previous studies could not verify the hypothesized double helix and did not elucidate the molecular configuration of double helical PBDT.

To compare with the nematic LC phase and discuss the origin of double helices in PBDT, we have studied a chemically similar control system, poly(2,2poly-2,2'-disulfonyl-4,4'-benzidine terephthalamide-disulfonyl-benzidine isophthalamide) (PBDI). PBDI is a non-LC polymer, and the only structural difference from PBDT is the meta linkage in the backbone instead of the para linkage. This relatively minor structural difference causes PBDI aqueous solutions (at any concentration) to exhibit no NMR, optical, or X-ray signatures of an LC phase[35]. Thus, the PBDI molecule is not a rigid rod and will not form the double helix. Thus, we demonstrate that the all-para linkage is essential to form the double helix structure.

In this article, we present complementary experimental and computational evidence for the PBDT double helix. Furthermore, we elaborate a number of similarities between PBDT and DNA molecules in order to highlight the unique characteristics of the double helical conformation of this synthetic rigid-rod polyelectrolyte. Probing the similarities between these two systems provides important insight towards the understanding of double helical systems and feeds into design principles for future discovery of functional materials.

## Results

**XRD and rigidity of the PBDT double helix**. The DNA double helical configuration was proposed by Watson and Crick[1,2] based on an XRD pattern of DNA fibers by Rosalind Franklin[36]. Here, we employ XRD to study the packing structure and morphology of PBDT aqueous solutions. Since it is difficult to obtain aligned PBDT fibers, we have directly run XRD experiments on concentrated and magnetically oriented PBDT aqueous solutions. Figure 1 summarizes the structural configuration of PBDT and our XRD results, including data and simulations. As shown in Fig. 1a, we observe a highly ordered XRD pattern from a 20 wt% PBDT aqueous solution after placing the sample in a **B** field (≥0.5 T). To eliminate the effect of the solvent $H_2O$, we subtract an intensity-scaled $H_2O$ diffraction pattern using MATLAB®. We include the relevant images in Supplementary Figure 1. Figure 1b shows the corresponding simulated results. The strong mono-domain orientational order maintained after removal of the sample from the **B** field indicates the extremely long persistence length of the PBDT chain[34], which we attribute to the rigid PBDT backbone and more importantly to the double helical structure.

Indeed, the aligned phase forms above a critical concentration (1.5 wt%)[34], where the persistence length (stiffness) along the PBDT rod axis[37] is >240 nm (Onsager theory)[38] and ~670 nm (Flory theory)[39]. See Supplementary Note 1 for additional details. This persistence length is substantially longer than DNA (~50 nm)[40], collagen triple helices (~52 nm)[41], and other rigid helical molecules such as poly(benzyl-l-glutamate) (PBLG)[42]. Thus, PBDT exhibits perhaps the highest known rigidity persistence length of any simple molecular structure. With refinements in synthesis, we obtain higher molecular weight PBDT (see SI for details). The aligned phase forms at concentrations down to 0.3 wt% PBDT, which represents a persistence length of >1.2 μm.

In Fig. 1a, the black arrow on the bottom right shows the direction of sample alignment in the instrument, which is also the aligned axis of molecular orientation. Alignment can be imposed by either a weak applied magnetic field or simply by the cylindrical confines of an X-ray capillary[34]. The red solid line that passes through the diagonal of the pattern is the meridian. The diffraction peaks on the meridian are called meridional reflections, which usually represent subunit axial translations[43]. The blue solid line perpendicular to the meridian is the equator, with peaks called equatorial reflections. A helical diffraction pattern is usually represented by a series of layers or arcs along the meridian, called layer lines[43,44]. The layer line with closest distance to the equator gives the pitch length for a single helix or half pitch length for a double helix such as that described here. Based on the relationship between real space and reciprocal space, the positions of the layer lines are related to successive orders of the pitch length ($P$), which in this case is 33.6 Å. Starting from the equator, for a double helix, the layer lines are at spacings $P/2$, $P/4$, and $P/6$[43,44]. All of the layer lines are at each side of the meridian, but usually not on the meridian, except for when the layer lines also correspond to the subunit translation[43,44]. Based on this

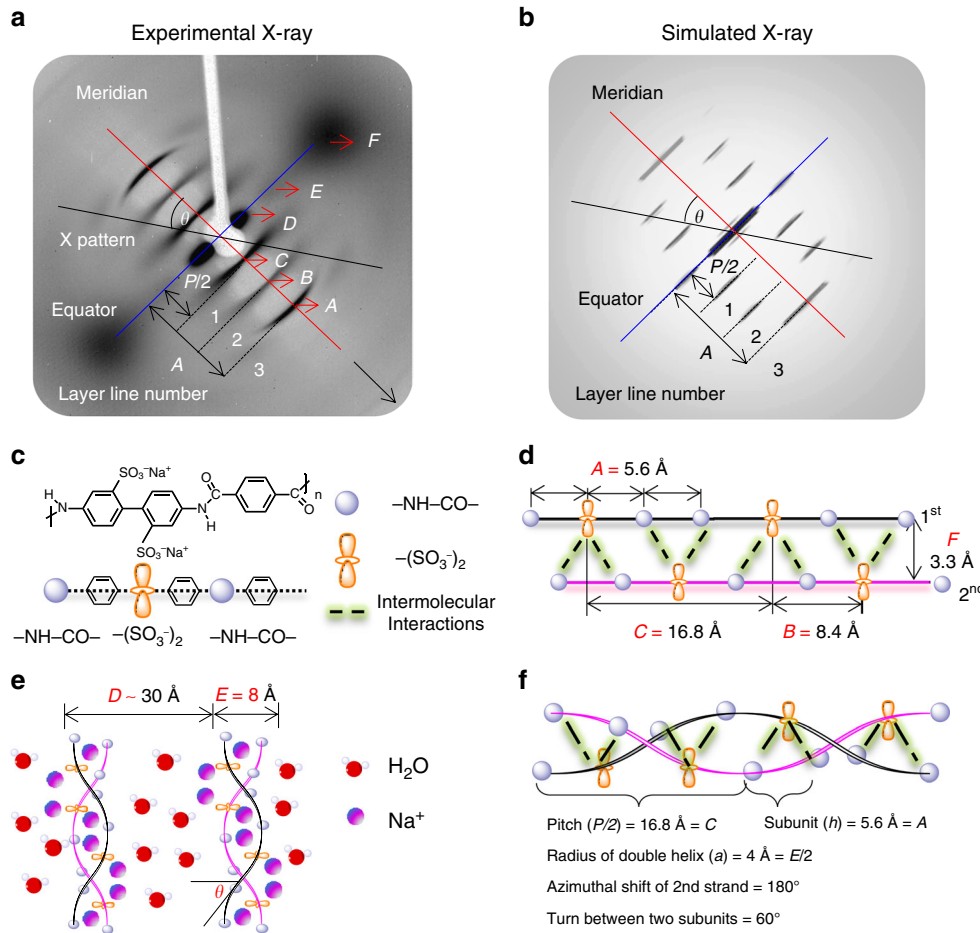

**Fig. 1** PBDT structural motifs along with experimental and simulated XRD results. **a** X-ray diffraction pattern for 20 wt% PBDT aqueous solution. The main diffractions are labeled A, B, C, D, E, and F and the helix tilt angle is θ. **b** The simulated X-ray diffraction pattern based on the HELIX software package (see also Supplementary Figure 1). The layer lines 1, 2, and 3 in the simulated results clearly mimic the experimental results. **c** The chemical repeat unit of PBDT includes one set of $-SO_3^-$ groups (two sulfonate groups from one biphenyl unit) and two $-NHCO-$ groups, each of which are mutually connected by one benzene ring. **d** The second PDBT strand is shifted 8.4 Å (B = P/4) away from the first strand along the helix axis. Numerous intermolecular interactions between chains (notably hydrogen bonding, dipole–dipole, and/or ion–dipole interactions between $-SO_3^-$ and $-NHCO-$ groups—shown as green dashed lines) and the rotation of each subunit contribute to the double helical conformation. **e** PBDT double helices self-assemble into an aligned (nematic) morphology in aqueous solution. The purple dots refer to $Na^+$ counterions. The red dots are water molecules. **f** Elucidation of interchain bonding and molecular packing for PBDT with the double helical conformation. The helical parameters used for the X-ray simulation are also listed (P = 33.6 Å, h = 5.6 Å, a = 4 Å, Azimuthal shift = 180°)

background, the main diffraction patterns in Fig. 1a are labeled with A, B, C, D, E, and F (A = 5.6 Å, B = 8.4 Å, C = 16.8 Å, D ≈ 30 Å, E = 8 Å, and F = 3.3 Å). By correlating the diffraction patterns and the helical parameters, we can elucidate the double helical conformation of PBDT chains in aqueous solutions as follows.

In Fig. 1c, the chemical repeat unit of PBDT includes a set of two $-SO_3^-$ groups that coincide approximately in a plane perpendicular to the molecular axis and two $-NHCO-$ groups that are mutually connected by one benzene ring. The theoretical chemical repeat unit length of PBDT along the average molecular alignment axis is ca. 17.4 Å[35] assuming all bonds are in the trans conformation. Here we observe a repeat distance along the aligned rod axis of 16.8 Å (C = P/2). This repeat unit is composed of three subunits with equal length of 5.6 Å (A = P/6). Figure 1d shows the combination of two PBDT chains, where the second strand is shifted 8.4 Å (B = P/4) away from the first strand. Intermolecular interactions such as hydrogen bonding (between the $-SO_3^-$ and $-NHCO-$ functionalities), π stacking, and dipole–dipole interactions along with the rotation of each subunit

contribute to the formation of the double helical conformation. B is the distance between adjacent $-SO_3^-$ functionalities along the polymer chain. The average inter-helix distance (within one double helix) is represented by F. Figure 1e shows schematically the self-assembly behavior of PBDT chains in $H_2O$ (red) solution with $Na^+$ counterions (purple). In 20 wt% PBDT aqueous solution, the rod–rod distance is ~30 Å, represented by D. This spacing is not clearly defined due to the interference of the diffracted beam with the beamstop, but we have further characterized this distance by SAXS, as introduced below. E corresponds to the overall average diameter of the double helix. In addition to the main diffraction peaks along the equator and meridian, we explain the displayed layer-line patterns by analogy with the layer-line diffractions of DNA fibers. The symmetry of a helical structure can usually be defined in terms of a number of parameters, including the subunit axial translation (h), the pitch of the helix (P), and the radius of the helix (r)[43,44]. The angle θ shown in Fig. 1e is the averaged angle from the perpendicular direction of the helix axis to the long (tangent) axis of the individual molecular chain, and can be extracted from the X-ray

experimental results. Distinct from $\theta$, the rotation angle for each subunit is 360°/6, where the denominator 6 is the number of subunits in one pitch length. To describe the origin of the layer-line patterns, we propose a model with critical parameters in Fig. 1f, based on which we simulate a diffraction pattern (Fig. 1b) using a helical diffraction simulation software package called HELIX[44]. We include detailed simulation parameters and information about HELIX in Supplementary Note 2 and Supplementary Figure 2. The simulated results show high coincidence with the experimental results. The main difference is the absence of reflections on the meridian along the layer line number 1 and 2. We attribute this discrepancy to two factors: (1) The HELIX software cannot differentiate the $-SO_3^-$ and $-NHCO-$ functionalities, which are represented as the same subunits in the simulation program. (2) The HELIX program also assumes a perfectly oriented chain, whereas in experimental X-ray, we have PBDT aqueous nematic liquid crystalline solutions. The imperfect alignment of the phase (orientational order parameter $S_{matrix} = 0.78$ as introduced later) results in significant smearing of these peaks (layer lines) into crescents[34]. The additional diffractions on the meridian with length of $B$ (8.4 Å) and $C$ (16.8 Å) (Fig. 1a) are both attributed to axial subunit translation of the $-SO_3^-$ functionalities between two chains and along one chain, respectively. The $B$ and $C$ features in particular show four-spot character (splittings on the layer lines) as observed in the simulations, but with smearing due to liquid crystalline orientational distributions of these rodlike molecules. Beyond these distortions, the layer-line spacings in the simulated results agree perfectly with the experimental results, thus confirming the double helical structure of PBDT.

**Alignment and counterion associations by NMR and SAXS.** In addition to XRD, quadrupolar NMR spectroscopy provides a complementary measurement of anisotropic structure and molecular dynamics in PBDT aqueous solutions. Strzelecka and Rill[45,46] have employed $^{23}$Na NMR to investigate concentrated sodium-DNA aqueous solutions. They observed an $^{23}$Na triplet spectrum, which they attributed to the interaction between the Na quadrupole moment and the Na$^+$ electric field gradient (efg) in the anisotropic ambient environment[45,46]. Herein, the quadrupolar splitting $\Delta\nu_Q$ of the $^{23}$Na nucleus in a uniaxially aligned system can be expressed as

$$\Delta\nu_Q = Q_p \rho S_{matrix} P_2(\cos\theta_Q) = Q_p \rho S_{matrix} \frac{(3\cos^2\theta_Q - 1)}{2}, \quad (1)$$

where $Q_p$ is the quadrupolar coupling parameter that represents the $^{23}$Na spectral splitting when the nucleus is static and perfectly aligned, and $\theta_Q$ is the averaged angle between the principle axis of the Na$^+$ efg tensor $V_{zz}$ and the alignment axis of the PBDT matrix[47]. $S_{matrix}$ is the order parameter of the aligned PBDT chain matrix, and $\rho$ is the scaling factor representing the interaction between the quadrupolar probe species $^{23}$Na and the host PBDT matrix[34,48,49]. The average angle $\theta_Q$ will be highly dependent on temperature, local asymmetric environment, and concentration of PBDT. The reported magnitude of the splitting $\Delta\nu_Q$ decreases with increasing PBDT concentration at low temperatures, whereas $\Delta\nu_Q$ increases with concentration at high temperature. These temperature- and concentration-dependent changes in $\Delta\nu_Q$ are consistent with the magnitude of $P_2(\cos\theta_Q)$. The quadrupole splitting and $P_2(\cos\theta_Q)$ converge to 0 when $\theta_Q$ is at the magic angle = 54.7°[45,46].

By developing the experimental design for PBDT solutions based on these two varying factors, (temperature and concentration), we observe that PBDT and DNA solutions display highly consistent NMR spectroscopic (quadrupole splitting) behaviors.

In Fig. 2a, the quadrupole splitting of $^{23}$Na will approximately converge to 0 at a null concentration $C_{PBDT} = C_0 \approx 10$ wt% at 25 °C. The rod–rod distance at this concentration is 41 Å (see Supplementary Note 3). We attribute this quadrupole splitting dependence to the fact that Na$^+$ ions can exchange, predominantly with $-SO_3^-$ groups, both along the individual charged rods (PBDT double helices) as well as between $-SO_3^-$ groups on different rods. In other words, Na$^+$ can jump between two rigid PBDT rods in an inter-helical interaction, or Na$^+$ can jump along a single rod in an intra-helical interaction. This process is both space and time averaged. Additionally, both intra-helical and inter-helical interactions are highly correlated to the concentration and temperature of PBDT aqueous solutions. As shown in Fig. 2b, at low PBDT concentration, the rod-rod distance is large, corresponding to dominant intra-helical Na$^+$ interactions (translations) along the polymer chain. Na$^+$ needs to axially translate 33.6 Å (one pitch length) to arrive at the next Na$^+$ with exactly the same coordination location. As concentration increases, the rod–rod distance decreases, and at the concentration 9.1 wt% we observe the intersection point (blue circle) of the experimental data shown in Fig. 2f with $\Delta\nu_Q = 0$ Hz. This specific point corresponds to isotropically averaged Na$^+$ interactions (inter-helical and intra-helical), as shown in Fig. 2c. With further concentration increase, we observe that $\Delta\nu_Q$ increases again, corresponding to the dominance of the inter-helical interaction as compared to the intra-helical interaction as shown in Fig. 2d. Based on this deduction, when $C_{PBDT} = 9.1$ wt% with $\theta_Q = 54.7°$, the ratio ($P/r$) of the pitch length (33.6 Å) to the rod–rod distance $r$ is $1/\sqrt{2}$. At this point where Na$^+$ experiences isotropic averaging, we observe $r = r_{isotropic} = 48$ Å, as displayed in Fig. 2c. In summary, compared to previous double-stranded DNA studies, the $^{23}$Na NMR of counterions[45,46] here shows splitting dependencies that provide more quantitative interpretation in terms of inter- and intra-strand ion exchange.

In order to further verify the proposed model, we employ SAXS to investigate the rod–rod distance in PBDT aqueous solutions with varying concentrations. As shown in Fig. 2e, we can access the rod–rod distance ($r$) based on the scattering vector $q$. By combining the SAXS results and our previously reported double-stranded hexagonal lattice model (Supplementary Note 3)[34], we use the relationship between rod–rod distance $r$ and polymer concentration $C_{PBDT}$ to obtain $r_{isotropic} = 43$ Å (at $C_{PBDT} = C_0 = 9.1$%), which agrees well with the $r_{isotropic} = 48$ Å obtained from the proposed model. Figure 2f shows the experimental $^{23}$Na splitting and $P_2(\cos\theta_Q)$ extracted from the double helix hexagonal lattice model. $\theta_Q$ can be obtained from Eq. (2).

$$\theta_Q = \arctan\left(\frac{r}{33.6}\right). \quad (2)$$

The blue area displays the concentration range where the inter- and intra-helical interactions are neutralized (isotropically averaged). Thus, in complement with X-ray scattering, we can utilize $^{23}$Na quadrupolar NMR spectroscopy to quantitatively verify the double helical structure of PBDT.

In addition to the concentration dependence of the $^{23}$Na NMR quadrupole splitting, we can also explore the $^{23}$Na quadrupole splitting as a function of temperature. We observe a similar splitting pattern dependence as with concentration, in which the splitting first decreases across the null point and then increases again with temperature (Fig. 2g). Furthermore, we observe that $C_0$ is proportional to temperature, as indicated by the red line at $\Delta\nu_Q = 0$ Hz. This unusual dependence is also clearly observed in aqueous solutions of DNA molecules[46]. The longitudinal ($T_1$) and transverse ($T_2$) spin relaxation times of Na$^+$ counterions in PBDT

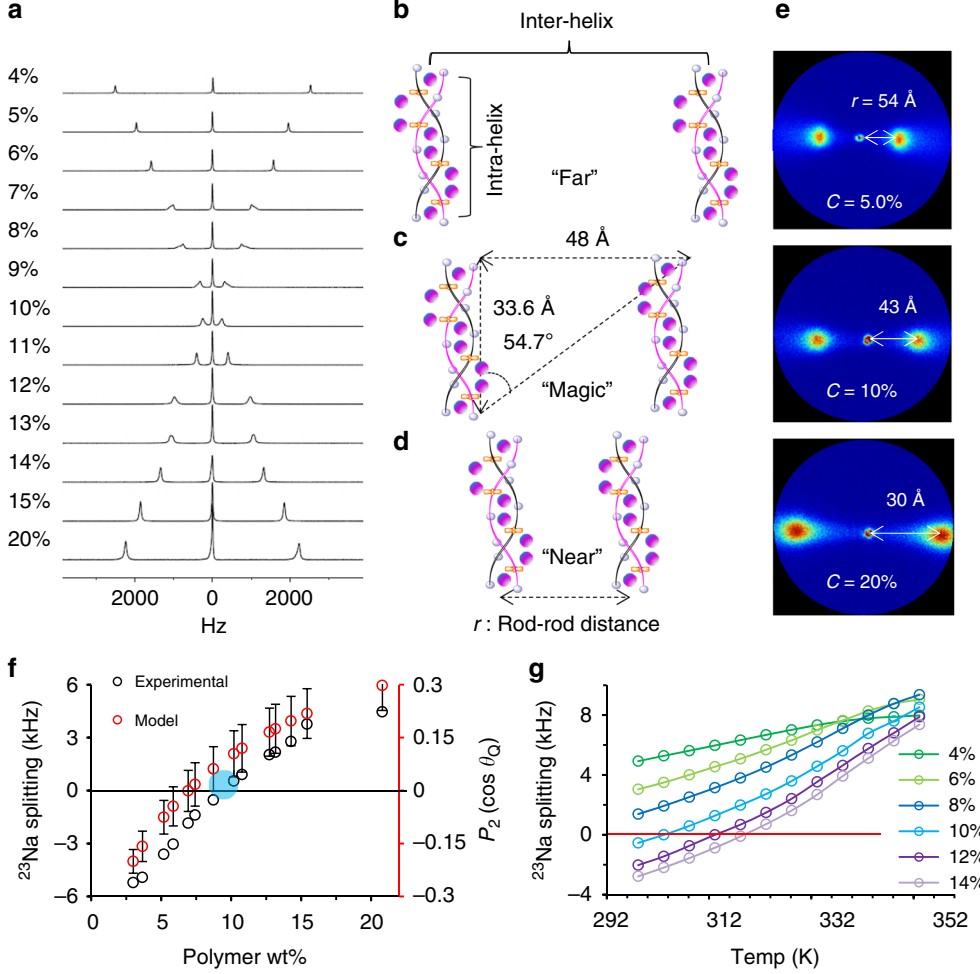

**Fig. 2** $^{23}$Na NMR spectroscopy and SAXS for PBDT aqueous solutions. Panel **a** shows concentration-dependent $^{23}$Na quadrupole spectra for PBDT solutions at 298 K. Panels **b**–**d** show configurations of PBDT self-assembly behavior with increasing concentration, and the critical geometric parameters at the null (isotropically averaged) point where Na$^+$ shows isotropic dynamics are displayed in **c**. Intra-helical Na$^+$ interactions dominate in **b**, the null point is at **c** and inter-helical Na$^+$ interactions dominate in **d**. Panel **e** shows SAXS results for PBDT aqueous solutions with $C_{PBDT} = 5\%$, 10%, and 20%. **f** The black open circles represent the observed $^{23}$Na quadrupole splitting as a function of increasing PBDT concentration. The red open circles show the calculated $P_2(\cos\theta_Q)$ based on a model incorporating the rod-rod distance ($r$) calculated from the double-stranded hexagonal lattice model and the pitch length ($P = 33.6$ Å) obtained from XRD. Panel **g** shows the $^{23}$Na quadrupole splittings as a function of temperature for PBDT with varying concentrations $C_{PBDT} = 4\%$, 6%, 8%, 10%, 12%, and 14%. The error bars for the $P_2(\cos\theta_Q)$ model points shown in **f** are dominated by the standard deviation in PBDT polymer weight percent ($C$) in solution (ca. 2.5%) as a percentage of the total concentration of polymer in each solution (e.g., 0.025 × 5 wt% polymer). $C$ determines the rod-rod distance ($r \sim C^{-0.5}$) and also $\cos\theta_Q$, where $\cos\theta_Q = P/(P^2 + r^2)^{1/2}$. Thus, the error bars are calculated based only on the error in PBDT concentration. Source data are provided as a Source Data file

solutions also parallel results for DNA, and are included in Supplementary Note 4 and Supplementary Figure 3.

**Molecular dynamics simulations of double helix formation**. Finally, we describe the PBDT chain configuration based on MD simulation studies conducted using particular simulation packages, thermodynamic conditions, and sets of force field parameters. As shown in Fig. 3, we investigate self-assembly of the double helical structure when two chains are present. A double helix structure is formed during 120 ns of simulation time in water at $T = 300$ K and pressure = 1 atm. The estimated pitch length for this model is 3–4 monomer lengths, which is somewhat longer than XRD results. We note that MD simulations also show stable double helix formation even when employing substantially different simulation conditions (NPT ensemble, NVT ensemble, in water using different force fields, and in vacuum—see Methods: MD simulations section and Supplementary Figures 4–11). In this case, simulations were performed using the Gromacs code

in the NPT ensemble[50]. Two PBDT monomers, each with four repeating units and measuring ≈6.8 nm in length, were initially placed side-by-side in a simulation box filled with water. After energy minimization, an equilibrium simulation was performed. During the first 30 ns, the two PBDT molecules intertwined with each other to form the double helical structure shown in Fig. 3a, e. The system was equilibrated for an additional 90 ns after the double helix self-assembled to ensure the structure is stable. Snapshots of the two PBDT monomers and their self-assembly process are shown in Fig. 3b–f. We performed two additional analyses to confirm that the double helical structure is stable and equilibrium was reached in our simulations (Supplementary Note 5, Supplementary Figures 4 and 5). Additionally, we computed the distribution of the Na$^+$ counterions around the sulfonate groups in the PBDT solution. The results are shown in Supplementary Figures 6a and 8. As a reference, the distribution of water molecules around the sulfonate groups is also shown in Supplementary Figure 6b.

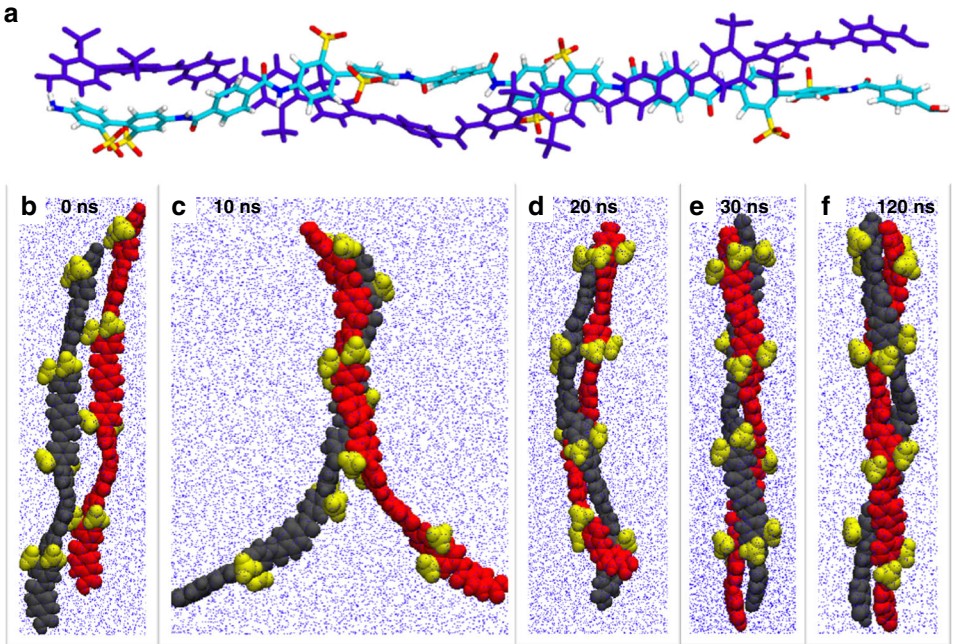

**Fig. 3** Molecular dynamics (MD) simulations of PBDT polymer chains in water. **a** Simulation result for two oligomers with four monomers run for 120 ns in water at $T = 300$ K and $P = 1$ atm. **b–f** Evolution of the conformation of the PBDT monomers during their self-assembly process. The snapshots show the two PBDT polyanions at **b** 0 ns, **c** 10 ns, **d** 20 ns, **e** 30 ns, **f** 120 ns in the simulation. Additional snapshots are shown in Supplementary Figures 6 and 8. The two PBDT monomers form a double helix structure at ≈30 ns, and the structure is stable for the rest of the simulation lasting 90 ns. The red and black balls denote the backbone atoms of the two PBDT monomers, and the yellow balls denote the sulfonate group atoms. The blue background dots denote the water molecules. Sodium counterions and explicit water atoms are omitted for clarity

Above all, we emphasize that the MD models confirm the existence of a double helical structure in this rigid-rod polyelectrolyte. It is clear that establishing this double helix relies on a wide variety of intermolecular interactions, most likely including ion–dipole, pi–pi stacking, hydrophobic effects, hydrogen bonding, and dipole–dipole interactions. This subtle self-assembly of a simple and compact molecular repeat unit involves more complex non-covalent interactions than the simple H-bonding found in DNA and yet with less molecular complexity than most other known double helical structures. Future studies will further unravel the specific interactions at play in order to provide additional insights for informed molecular design for tailored self-assemblies.

## Discussion

In this communication, we describe the molecular configuration of the sulfonated aramid PBDT in aqueous solutions. Based on complementary characterizations, we find a number of similarities between PBDT and DNA molecules and demonstrate the unique double-stranded helical structure of PBDT chains in aqueous solution. The strong self-assembly and extreme rigidity present in the PBDT system arises from a varied and synergistic array of interchain interactions, which are more complex and most likely stronger than the simple H-bonding-driven double helical self-assembly mechanism for DNA. We have tried many solvents to fully dissolve or denature PBDT and break the double helix, including acetone, toluene, alcohols, THF, ethylene glycol, several salt solutions at varying concentration, and DMSO. However, we found that none of them can dissolve PBDT or break the double helix into single chains. From the perspective of macromolecular structural and morphological understanding, this work provides critical ideas and directions for comprehending molecular packing and ionic association behavior in

rigid-rod polyelectrolytes and other self-assembled structures. Furthermore, this work not only resolves the configuration of the PBDT molecular chain but also lays the foundation for discovering broader applications based on polyelectrolytes with extreme axial rigidity, such as highly conductive yet stiff molecular ionic composites[32]. Finally, this highly charged and rigid double helical structure (and/or its derivatives) could potentially become a new one-dimensional (1D) material additive for more general incorporation into functional composites.

## Methods

**PBDT solutions.** The synthesized PBDT polymer was dried in a vacuum oven at room temperature for 2 days. PBDT aqueous solutions with weight percentages (wt%) ranging from 4% to 20% were prepared by loading the quantified amount of polymer (PBDT) and solvent ($D_2O$) into 5 mm NMR tubes. $D_2O$ was used as received from Cambridge Isotope Laboratories (~99.9%). To prevent evaporation of water, all of the samples prepared in NMR tubes were flame-sealed immediately after loading. In addition, to ensure complete dissolution and homogenization of the polymer in water, the sealed NMR tubes were equilibrated at 80 °C in a water bath for 1 week prior to X-ray and NMR measurements.

**[23]Na NMR.** The Bruker Avance III WB 400 MHz (9.4 T) NMR spectrometer was equipped with a 5 mm axial saddle [23]Na rf coil. A simple pulse-acquire sequence with a 90° RF pulse of 5 μs was applied for all measurements from 25–80 °C.

**X-ray diffraction.** The XRD measurements were performed with a single-crystal diffractometer (Rigaku Oxford Diffraction Xcalibur Nova) equipped with an Onyx CCD detector and a Cu microsource. The setup was operated at 49.5 kV and 80 mA at room temperature. The 20% PBDT aqueous solution was sealed in a quartz capillary with 1.5 mm diameter and 0.01 mm wall thickness and mounted on the edge of a steel pin. The sample-to-detector distance was set to 120 mm, thus offering a scattering angle $2\theta_s$ from 5° to 34°. The capillary was oriented 45° angle relative to the beamstop thus minimizing interference with the beamstop in the diffraction patterns. The sample was rotated in steps of 2° in phi (along the fiber axis). To increase the signal-to-noise ratio of the diffraction pattern, a total of 200 frames, each with 1200 s exposure time were summed. The XRD patterns were

collected and analyzed with CrysAlisPro (v1.171.37.35, Rigaku Oxford Diffraction, 2015; Rigaku Corporation, Oxford, UK).

**Small-angle X-ray scattering**. All SAXS results were acquired using a Rigaku S-Max 3000 pinhole SAXS system. An X-ray beam with a wavelength $\lambda$ of 0.154 nm (Cu K$\alpha$) was obtained by using a copper rotating anode. The sample-to-detector distance was set to 1600 mm. A silver behenate standard sample was used to calibrate the relationship between pixel position and scattering vector $q$. PBDT aqueous solutions with specific concentrations were sealed in the same kind of capillary as introduced in XRD section. Ahead of the SAXS measurements, the capillaries were loaded with solution samples and placed axially along a 7.1 T magnetic field to achieve uniaxial alignment. During the SAXS experiments, the capillaries were placed horizontally in the SAXS sample chamber without a magnetic field. The SAXSGUI software package (Rigaku Innovative Technologies, Inc.) was used to generate the integrated SAXS intensity $I(q)$ as a function of scattering vector $q$, where $q = (4\pi \sin\theta_s)/\lambda$ and $\theta_s$ refers to one-half of the total scattering angle.

**MD simulations**. MD simulations were performed to investigate the self-assembly of two initially separated PBDT oligomers in aqueous solution. Two PBDT monomers, each with four repeating units and measuring ~6.8 nm in length, were initially placed side-by-side in a $6 \times 6 \times 10$ nm$^3$ simulation box. The simulation box was filled with 11,850 water molecules using the Packmol code[51]. Sixteen Na$^+$ ions were included to ensure the electro-neutrality of the system. The simulation box was periodic in all three directions. The force field parameters for the PBDT polyanions were generated using the Swissparam server[52]. The TIP3P model was employed for the water molecules[53]. The force fields for the sodium ions were taken from the work by Joung and Cheatham[54].

Simulations were performed using the MD code Gromacs 4.5 (ref. [50]). First, an energy minimization was conducted using the steepest descent method, and the minimization was terminated when the maximal force in the system became smaller than 1000.0 kJ mol$^{-1}$ nm$^{-1}$. Next, a 120 ns equilibrium simulation was performed in the NPT ensemble using a time step size of 2 fs. The non-electrostatic interactions were computed via direct summation with a cut-off length of 1.2 nm. The electrostatic interactions were computed using the Particle Mesh Ewald (PME) method[55]. The real space cut-off and FFT spacing were 1.2 and 0.12 nm, respectively. All bonds were constrained using the LINCS algorithm[56]. The system temperature was maintained at 300 K using the Nose-Hoover thermostat (time constant: 1 ps)[57,58] and the pressure was maintained at 1 atm using the Parrinello-Rahman barostat (time constant: 10 ps)[59,60]. The trajectory was saved every 10 ps and analyzed using the tools provided by Gromacs.

## Data availability

All relevant data generated during and/or analyzed during the current study are available from the corresponding author upon reasonable request. In addition, MD simulation atom coordinate data (GROMACS gro format) and X-ray diffraction raw (img) datasets are available on the MaterialsCloud data archive, https://archive.materialscloud.org/2019.0005/v1. No data in this manuscript are restricted in terms of availability.

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

## Acknowledgements

The authors wish to thank Professors Dimitri Ivanov and Edward T. Samulski for helpful discussions. This work was supported in part by the National Science Foundation under award numbers DMR 1507764, 1810194, and 1507245. Any opinions, findings, and conclusions or recommendations expressed in this material are those of the author(s) and do not necessarily reflect the views of the National Science Foundation (NSF). This work was also supported in part by the Institute for Critical Technology and Applied Science (ICTAS) and the Open Access Subvention Fund (OASF) at Virginia Tech.

## Author contributions

Y.W., C.S., G.B.F. and R.B.M. designed and conducted X-ray experiments and simulations. Y.W., C.J.Z. and L.A.M. conducted NMR experiments and developed related models. J.G. and M.H. synthesized the materials. S.T.B. and B.E. conducted MD simulations in vacuum. Y.H., Z.Y. and R. Q. conducted MD simulations in water. Y.W. and L.A.M. developed and compiled concepts and wrote the main paper. Y.H., Z.Y., J.G., S.T.B., C.S., G.B.F., C.J.Z., M.H., R.B.M., B.E., T.J.D. and R.Q. assisted with compiling and editing the paper.

## Additional information

**Competing interests:** The authors declare no competing interests.

