## [Peer Review File · Nature Communications]

Reviewers' comments:

Reviewer #1 (Remarks to the Author):

This manuscript reports on the discovery that a previously described rigid poly-electrolyte derived from a polyarylamide (PBDT) in fact exists as a double helix in solution. Although there are now quite a few non-natural organic backbones known to form multistranded helices, I find this result particularly important in that the polyaramide studied here and its self-assembling mode do not relate to any other structure. The manuscript is overall quite convincing and well presented. It is interesting also that this contribution comes from polymer scientists whereas most of the published work on synthetic multistranded helices have been developed by supramolecular and organic chemists. The encounter between these communities is likely to stimulate the field. I thus think there is excellent ground for a Nat Comm article but also that some complements and clarifications are needed.

1°) The abstract and the first part of the introduction do not reflect a thorough knowledge of the literature on artificial double helix formation. Citations and statements are not all appropriate.

- For instance, artificial multistranded organic helices are not so few anymore: besides amidinium-carboxylate oligomers, Yashima (Univ. Nagoya) has described oligoresorcinols and also boronate linked duplexes, Huc (Univ. Bordeaux) has reported a number of duplexes and triplexes from aromatic aryl amides, James Wisner (Univ. Western Ontario) has described hydrogen bonded double helices, Gopi (IISER Pune) has recently presented double helices from gamma peptides, and synthetic variants of Gramicidin D have also been shown to form duplexes. The authors should emphasize in what their structure differs from those above (e.g. the para-substitution gives rises to a large helix pitch in contrast with the many meta-substituted aromatics described before that are more curved).
- It is a misconception that chirality is necessary to form double helices, no originality claim of that kind should appear. Most of the system above (Yashima's resorcinols, Huc, Gopi, Wisner, it-PMMA) have no chirality, and when present, chirality is often not necessary: achiral PNA forms racemic double helices with itself, Yashima's carboxylate-amidinium oligomers also hybridize in the absence of chirality.

2°) The comparison of PBDT with other double helices like DNA is relevant in terms of the contribution of multiple strands to stiffness and persistent length.

- How does collagen fares in this comparison?
- Why does PBDT not form gels like collagen?
- Please provide an estimate of the strength and thermodynamics of PBDT double helix formation? Is there an entropic component, i.e. melting at high temperature like DNA and collagen?
- How accurate are chain lengths measurements (>180kg/mol !!)? Can't the chains be a succession of shorter strands linked as double helices through dangling ends and only an apparent molecular weight is measured? If chain length measurement is not accurate, what is the consequence on the estimated persistent length: can one determine persistent length actually longer than the chains, or is that persistent length only apparent?

3°) I am skeptical about the conclusions drawn from 100 ps long MD simulations for phenomena that probably take much longer time. Along the same line, why look at a polyelectrolyte in vacuum?

Reviewer #2 (Remarks to the Author):

Wang et al. report an interesting system based on the aromatic polymer (PBDT), which appears to form a double helix resembling DNA. This work has the potential to be of interest to a broad audience and of great importance to the community, but there are a number of issues that should

be addressed by the authors prior to publication in any journal:

The Introduction should more clearly spell out what work has been done before and what is novel in the current work. For example, the first sentence on page 3 (beginning "Herein, we describe a unique aromatic sulfonated polyamide...") suggests to the reader that this is a new molecular system. In fact, this particular molecule has been the subject of a number of papers from this group and others (references 23-28). One of these papers from 2014 even hypothesizes a double helical structure based on SAXS and the lattice model. The introduction should include more discussion of previous structural characterization beyond the sentence "We have described a nematic LC model of PBDT aqueous solutions previously, but the detailed molecular configuration of PBDT has not been fully investigated."

X-ray Diffraction

Figure 1 shows really beautiful X-ray diffraction results. However, the data is quite rich and the reader would benefit from additional discussion for this dense data set. The patterns of B and C particularly need further explanation. For example, why is B not strongly split as it is in the simulated X-ray? Is P/4 on the meridian? Also, how is theta calculated—is this simply $360^\circ/6$ or is this obtained from the experimental values for P/2?

NMR

The sodium-23 NMR is a very nice result, but it is not entirely clear how this relates to the central story of the double helix? Is there some aspect of the rates of intra-helix exchange that would further support the double-stranded model? Maybe it would be helpful to discuss how these observations compare to the analogous previously reported NMR for dsDNA. What is the source of the asymmetry in the peaks at 7% and 8% in Figure 2a? How are the error bars determined for model sodium-23 splitting in Figure 2f? Are those related to the mean square error = 0.078 reported in the Supplementary Information?

This paper would be greatly strengthened by comparison to a control system that does not form the double helix. I realize this could be quite challenging, but might be possible with a different molecular structure (ester or methylated amide), but then the polymer might be different in other ways, such as molecular weight. Is it possible to suppress the double helix formation with a different solvent or with a different counterion? It would also be helpful to confirm that suppressing the double helix reduces the persistence length.

MD Simulations

While the results of the simulations look good and the final structures seem to support authors' claims, many details, analysis, and results are missing. In particular, the paper is missing key details about the systems: box size, solvent model and ions added for the first simulations, time steps for both; and the information actually included is spread between the results and the methods sections. While there are some comments in the Results section, the complete details must be added to the Methods section. Many of the methods are missing references (TIP3P water model, AMBER force field, and CM3D software).

The authors mention on line 249 that they will investigate self-assembly "when two or more chains are present", but only two are shown. They also mention a set of initial simulations (line 251) but it looks like they just show one. They should discuss whether the results have been replicated in multiple simulations or not, and if so comment on the level of reproducibility.

How was the optimized structure of PBDT obtained? Was energy minimization done before running any of the simulations? The Methods section should include these details.

For the first simulations, 100 ps looks quite short, but they do not justify why this is sufficient. Also, they comment again about the three chains, but the results are not shown. The SI should include the initial configurations that show up at the beginning and a description of how long the

helix configuration lasts.

Regarding the second method used, I do not understand the sentence: "The estimated pitch length for this model is 3-4 monomer lengths, which is somewhat longer than the first simulation." Are they trying to justify the difference in length for the systems in the two simulations? They say these simulations are also NVT. Is this correct? In solution, NPT is typically used. Do the different thermodynamic conditions mean the presence of water? If these were actually carried out in NVT ensemble, an explanation for why should be given.

Here they say "after energy minimization", but the method should be specified. Then they say "the system was equilibrated", but it is not clear if they refer to a simulation previous to the results shown or if they call equilibration the simulation shown here. Regardless, the pictures in Figure S3 look the same, but a quantitative measure of equilibration must also be shown, such as RMSD, some geometric parameter (like distances between groups that match the experimental data) or SASA, as a function of time. If the system is really equilibrated the plot should reach a plateau. This kind of analysis should also be done for the first system to justify that the simulation time was sufficiently long. I think that another interesting measurement is the extension of the molecules with time.

These analyses are quite standard, and I am surprised they do not include any, all they did was measuring the distances that matched the experiments but, as I said, this is totally cherry picking, maybe that is the only frame of the simulation where they have that distance. Also, they are quite precise in giving the number 8.4 Å, but it is not specified how this is obtained (perhaps center of mass to center of mass?).

SI Figures S3 and S4 show snapshots for the simulation in solution. Figure S3 appears to show the initial and 30 ns data reported in Figure S4. If so, it seems redundant to have both figures, unless Figure S3 is meant to emphasize the organization of water and sodium ions. The main text (line 269 and the legend Figure S4) state "The system was further equilibrated for ~ 90 ns to ensure the structure is stable." This could be made clearer by saying "for an additional 90 ns..." since the total simulation time labeled in Figure S4 was 120 ns.

Reviewer #3 (Remarks to the Author):

This is an interesting paper, reporting the a double helix macromolecule with a rigid and simple molecular structure. As I am a computational biophysicist, I address here only general points of the paper as well as the MD simulations:

1. Some sentences are redundant and obvious. For instance:

„X-ray diffraction (XRD) is commonly used to determine the crystalline or semi-crystalline structure of small molecules, proteins and macromolecules“.

Please remove them.

2. Some sentences can be shortened without any loss of information. For instance, replace "In 1953, the DNA double helical configuration was first proposed by Watson and Crick^{1,2} based on an XRD pattern of DNA fibers by Rosalind Franklin. In 1953, the DNA double helical configuration was first proposed by Watson and Crick^{1,2} based on an XRD pattern of DNA fibers by Rosalind Franklin.²⁹ Following a similar approach, we also employ XRD to study the packing structure and morphology of PBDT aqueous solutions." with "Following Watson and Crick's approach for the DNA double helix (1,2), we employ XRD to study the packing structure and morphology of PBDT aqueous solutions" .

3. Can the authors write a modified version of the HELIX program so that it can differentiate the –SO₃ - and NHCO– functionalities? This would address an important point of the paper.

4. As for the MD simulations, the authors should report only the results for the simulation in water. This referee is highly skeptical that the authors can obtain any reliable information for their anionic polymer by performing simulations in vacuo. This has been amply demonstrated for DNA double helix simulations.

5. In the MD simulations in water:

- What is the pressure?

- Why didn't the authors perform simulation in the canonical ensemble?

-Can the authors describe the counterions distribution quantitatively ?

-What is the rationale based on which they used the OPLS force field? Has this force field already been tested for systems similar to that used here?

-What was the time step used?

-How did the authors assess quantitatively that the structure was equilibrated (see SI, "In Figure S3, after energy minimization, the system was equilibrated")?

Notes: Reviewers' comments and text in the original manuscript are shown in **black**, while author responses and new text are shown in **blue**. All line numbers stated regarding the locations of modified text are from the original manuscript PDF.

Response to Reviewer 1's Comments

Statement: *This manuscript reports on the discovery that a previously described rigid poly-electrolyte derived from a polyarylamide (PBDT) in fact exists as a double helix in solution. Although there are now quite a few non-natural organic backbones known to form multistranded helices, I find this result particularly important in that the polyaramide studied here and its self-assembling mode do not relate to any other structure. The manuscript is overall quite convincing and well presented. It is interesting also that this contribution comes from polymer scientists whereas most of the published work on synthetic multistranded helices have been developed by supramolecular and organic chemists. The encounter between these communities is likely to stimulate the field. I thus think there is excellent ground for a Nat Comm article but also that some complements and clarifications are needed.*

Response: We thank the reviewer for the succinct summary of our work and for recognizing its potential importance. We have revised the manuscript carefully based on these comments.

Comment #1: *The abstract and the first part of the introduction do not reflect a thorough knowledge of the literature on artificial double helix formation. Citations and statements are not all appropriate.*

- *For instance, artificial multistranded organic helices are not so few anymore: besides amidinium-carboxylate oligomers, Yashima (Univ. Nagoya) has described oligoresorcinols and also boronate linked duplexes, Huc (Univ. Bordeaux) has reported a number of duplexes and triplexes from aromatic aryl amides, James Wisner (Univ. Western Ontario) has described hydrogen bonded double helices, Gopi (IISER Pune) has recently presented double helices from gamma peptides, and synthetic variants of Gramicidin D have also been shown to form duplexes. The authors should emphasize in what their structure differs from those above (e.g. the para-substitution gives rises to a large helix pitch in contrast with the many meta-substituted aromatics described before that are more curved).*

- *It is a misconception that chirality is necessary to form double helices, no originality claim of that kind should appear. Most of the system above (Yashima's resorcinols, Huc, Gopi, Wisner, it-PMMA) have no chirality, and when present, chirality is often not necessary: achiral PNA forms racemic double helices with itself, Yashima's carboxylate-amidinium oligomers also hybridize in the absence of chirality.*

Response: We thank reviewer for this suggestion and for pointing out these previous related works. To relate the previous work on artificial multi-stranded organic helices and the structural differences in PBDT, we have added further references in our paper, including works from Yashima, Huc, Gopi et al. We also emphasize the importance and uniqueness of *all-para* structures of PBDT in forming the double helix structure. In addition, as far as we know, PBDT is the first instance of synthetic polymer that forms a double helix in aqueous solution without regard to some smaller oligomeric systems. In terms of chirality, we agree with the reviewer that chirality is not

necessary to form double helices, and that this already has precedent in the literature. Thus, we have removed the statement in the paper about chirality previously being necessary to contribute to the double helix structure.

We have modified the text as follows:

Page 2, line 43: Added “To the best of our knowledge, with the exceptions of some oligomers e.g. Yashima’s oligoresorcinol²⁵ and Huc’s heteromeric oligoamides²⁶, no synthetic motifs have been demonstrated to form double helices in water.”

Page 3, line 52: Added “As far as we know, this is the first instance of a synthetic polymer that can form a double helix in aqueous solutions with the exception of the oligomers mentioned above and isotactic poly(methyl methacrylate) (*it*-PMMA),²⁷ which only forms double helices in the solid crystalline state.”

Page 2, line 46: Deleted “Double helix formation in aromatic oligoamides is driven by incorporation of a chiral center or through chiral solvation.”

Comment #2: *The comparison of PBDT with other double helices like DNA is relevant in terms of the contribution of multiple strands to stiffness and persistent length.*

- How does collagen fares in this comparison?

Response: Here we compare the stiffness and persistence length of collagen based on the reports by Vesentini et al (*Ligaments and Tendons Journal 2013, 3, 23*). As we know, collagen forms a triple helix. Assuming a cylindrical geometry, and considering a collagen molecular radius of 0.8 nm, the Young’s modulus of individual molecular strand is obtained equal to 4.62 ± 0.41 GPa. The value of the persistence length $L_p = 51.5 \pm 6.7$ nm is very close to that of DNA molecules. In contrast, the PBDT double helices introduced in the present paper show a persistence length of ~ 1 micrometer with a molecular weight of 20 to ~ 100 kg/mol based on the molecular weight determinations and Onsager and Flory theories introduced in the SI.

We have modified the text as follows:

Page 4, line 88: Added “...and collagen triple helices (~ 52 nm)³⁹” after “This persistence length is substantially longer than DNA (~ 50 nm),³⁸”

- Why does PBDT not form gels like collagen?

Response: PBDT forms gels based on specific ion exchange processes with specific solvents, e.g., alkali solutions and many ionic liquids (C₂mimBF₄), and these systems have been introduced in previous reports.

Additionally, in the present manuscript, we have mentioned that PBDT can form gels.

Page 3, line 56: “This synthetic sulfonated aramid polyanion, poly-2,2’ -disulfonyl-4,4’ -benzidine terephthalamide (PBDT), can be used to form a unique series of hydrogels and ion gels, which have displayed great potential as next-generation functional materials for batteries, fuel cells and optical sensors.”

As reported previously by Gong *et al.*, hydrogels with a fibrillar structure based on PBDT aqueous solutions can be easily obtained by exchanging PBDT aqueous solution with NaOH aqueous solution.

Regarding collagen, at the molecular level collagen is a triple helix of polypeptide chains. At slightly above room temperature, individual collagen molecules assemble hierarchically into fibrils, with highly ordered structures of 100 nm in diameter and tens of micrometers in length. Finally, these fibrils will self-assemble into networks. The collagen rod-shaped molecule (or tropocollagen) is a subunit of larger collagen fibril aggregates. The subunit is made up of three polypeptide chains. These three left-handed helices are twisted together into a right-handed coiled coil, being stabilized by numerous hydrogen bonds and intra-molecular van de Waals interactions as well as some covalent bonds, and further associated into right-handed microfibrils (~40 nm in diameter) and fibrils (100-200 nm in diameter) with unusual strength and stability (Book: Gorgieva, S. & Kokol, V. *Collagen- Vs. Gelatine-Based Biomaterials and Their Biocompatibility: Review and Perspectives*).

Thus, PBDT gels show substantially different structures in gels than collagen, although some molecular interaction aspects (helices, fibrils) are similar, especially in the hydrogel systems.

- Please provide an estimate of the strength and thermodynamics of PBDT double helix formation? Is there an entropic component, i.e. melting at high temperature like DNA and collagen?

Response: We thank reviewer for these significant questions and comments. PBDT forms a nematic LC phase in water above a critical concentration of ~ 1 wt%. We propose that a wealth of intermolecular interactions contribute to the formation of PBDT double helix, including hydrogen-bonding, hydrophobic effects, dipole-dipole interactions, ion-dipole interactions, and inter-strand van der Waals forces. It is impossible to melt the PBDT molecule, since the melting temperature of PBDT is higher than the boiling point of water, and (in the solid state or ion gel state) the degradation temperature of PBDT. Thus, we have tried heating the solution and changing to many different solvents and denaturing agents, but we have found no means yet to disassemble the double helix of PBDT. The possible exception is dissolution in fuming sulfuric acid, which is cumbersome, and which in earlier studies yielded a GPC measurement of single-stranded polymer molecular weight (See reference 33, *Macromolecules* 47, 2984, 2014). Thus, the double helix structure is extremely stable. Though at this time we cannot quantify the strength and thermodynamics of PBDT double helix formation, the MD simulation generates the double helix structure based on a thermodynamic study of the system. Indeed, every set of MD simulation force field parameters we have tried (4 so far, not all details included here) and under conditions of both water solution and vacuum, we observe self-assembly of the double helix. As we mentioned in the manuscript, *page 15, line 278*, “Above all, we emphasize that the MD models confirm the existence of a double helical structure in this rigid-rod polyelectrolyte. It is clear that establishing this double helix relies on a wide variety of intermolecular interactions, most likely including ion-dipole, pi-pi stacking, hydrophobic effects, hydrogen bonding, and dipole-dipole interactions.” Please also refer to our *Response to Comment 4 of Reviewer 2* and the corresponding revisions we added.

- How accurate are chain lengths measurements (>180kg/mol !!)? Can't the chains be a succession of shorter strands linked as double helices through dangling ends and only an apparent molecular weight is measured? If chain length measurement is not accurate, what is the

consequence on the estimated persistent length: can one determine persistent length actually longer than the chains, or is that persistent length only apparent?

Response: We thank the reviewer for raising this question. It is known that determining molecular weight even for conventional (flexible) charged polymers is difficult, and this determination is *also* difficult for liquid crystalline polymers with no charges. There is simply no reliable way to determine MW. However, we have estimates based on gel permeation chromatography in fuming sulfuric acid, which is what is used to determine MW for Kevlar. The persistence lengths we observe based on the critical concentration for nematic phase formation are based on established theories, and for Onsager nematic LCs (the basis for this rigid-rod persistence length theory) one expects orientational order parameters of close to 0.8, which is exactly what we observe for PBDT in solution. We cannot be sure at this point whether the persistence length is longer than individual polymer chains, and this will form the basis for future studies and publications. It is certainly possible (and quite probable, we expect) that the double helices can be a succession of shorter strands end-linked into longer double helices. We mentioned exactly this possibility in the *Supplementary Information, page 2, line 43*, "...Indeed, L_p may encompass multiple individual polymer chains that entwine (or interleave) axially to form longer double helices than can be achieved with a simple combination of two chains." Unraveling this mystery is indeed an active area of study in our group.

We have modified the text as follows:

Page 4, line 91: Deleted "of $M_w \approx 180$ kg/mol"

Comment #3: *I am skeptical about the conclusions drawn from 100 ps long MD simulations for phenomena that probably take much longer time. Along the same line, why look at a polyelectrolyte in vacuum?*

Response: We thank the reviewer for raising this question. These vacuum simulations were preliminary to our aqueous simulations, and indeed show double helix formation at different temperatures and at relatively short simulation times (~ 100 ps), and these simulations maintain the stable double helix structure up to 300 ps. However, we agree with the reviewer that the simulation results in vacuum are not nearly as relevant or powerful as the simulations in water. Thus, we have moved all of the vacuum simulation results to *Supplementary Information* (Figure S6).

Response to Reviewer 2's Comments

Statement: Wang *et al.* report an interesting system based on the aromatic polymer (PBDT), which appears to form a double helix resembling DNA. This work has the potential to be of interest to a broad audience and of great importance to the community, but there are a number of issues that should be addressed by the authors prior to publication in any journal.

Response: We thank the reviewer for the succinct summary of our work and for recognizing its potential importance. Following the reviewer's constructive suggestions and comments, we have revised the manuscript carefully. Below please find our responses to these suggestions and comments.

Comment #1: *The Introduction should more clearly spell out what work has been done before and what is novel in the current work. For example, the first sentence on page 3 (beginning "Herein, we describe a unique aromatic sulfonated polyamide...") suggests to the reader that this is a new molecular system. In fact, this particular molecule has been the subject of a number of papers from this group and others (references 23-28). One of these papers from 2014 even hypothesizes a double helical structure based on SAXS and the lattice model. The introduction should include more discussion of previous structural characterization beyond the sentence "We have described a nematic LC model of PBDT aqueous solutions previously, but the detailed molecular configuration of PBDT has not been fully investigated."*

Response: We thank reviewer for this significant question and comments. We have incorporated additional literature into our paper and made some corrections to the introduction. We also emphasize the importance and uniqueness of all-para structures in formation of the double helix structure of PBDT. Please refer also to our response to Comment #1 of Reviewer 1.

As the reviewer mentions, PBDT is not a newly synthesized polymer and it has been investigated in many previous papers. However, in this paper we describe in significant detail the double helix model based on extensive evidence and a unique combination of characterization and simulation methods. In our previous paper, we only proposed the possibility of a double stranded nematic LC model in PBDT, and the present paper introduces more detailed experimental and theoretical results to validate and expand on our previous suggestion. In addition, we expect the methodology in the present work can be extended for characterization of other multi-stranded molecules in liquid crystalline systems.

We have modified the text as follows:

Page 3, line 62: Deleted "We have described a nematic LC model of PBDT aqueous solutions previously, but the detailed molecular configuration of PBDT has not been fully investigated."

Page 3, line 62: Added "Using ^2H NMR spectroscopy and small-angle X-ray scattering, we previously investigated nematic LC ordering and polymer chain-chain distance in PBDT aqueous solutions.³⁰ However, our previous studies could not verify the hypothesized double helix and did not elucidate the molecular configuration of double helical PBDT."

Comment #2: X-ray Diffraction

Figure 1 shows really beautiful X-ray diffraction results. However, the data is quite rich and the reader would benefit from additional discussion for this dense data set. The patterns of B and C particularly need further explanation. For example, why is B not strongly split as it is in the simulated X-ray? Is P/4 on the meridian? Also, how is theta calculated—is this simply $360^\circ/6$ or is this obtained from the experimental values for P/2

Response: We thank reviewer for the significant questions and comments. Yes, both P/2 and P/4 are on the meridian, which are the axial subunit translations. We have explained two reasons for the disturbance or modification of these two diffraction features in the paper.

We have modified the text as follows:

Page 8, lines 159: Modified “(2) The HELIX program also assumes a perfectly oriented chain, whereas in experimental X-ray, we have PBDT aqueous *nematic liquid crystalline* solutions. The imperfect alignment of the phase (orientational order parameter $S = 0.78$) results in significant smearing of these peaks (layer lines) into crescents.”

Page 8, lines 150: Added “The angle θ shown in **Figure 1e** is the averaged angle from the perpendicular direction of the helix axis to the long (tangent) axis of the individual molecular chain, and can be extracted from the X-ray experimental results. Distinct from θ , the rotation angle for each subunit is $360^\circ/6$, where the denominator 6 is the number of subunits in one pitch length.”

Page 8, lines 164: Added “The B and C features in particular show four-spot character (splittings on the layer lines) as observed in the simulations, but with smearing due to liquid crystalline orientational distributions of these rodlike molecules.”

Comment #3: NMR

The sodium-23 NMR is a very nice result, but it is not entirely clear how this relates to the central story of the double helix? Is there some aspect of the rates of intra-helix exchange that would further support the double-stranded model? Maybe it would be helpful to discuss how these observations compare to the analogous previously reported NMR for dsDNA. What is the source of the asymmetry in the peaks at 7% and 8% in Figure 2a? How are the error bars determined for model sodium-23 splitting in Figure 2f? Are those related to the mean square error = 0.078 reported in the Supplementary Information?

Response: We thank reviewer for the insightful questions and comments. We are aware of no previous reports about the idea of specific inter-strand and intra-strand exchange of counterions such as Na^+ in double helix systems. In dsDNA, previous ^{23}Na NMR of counterions (references 43 and 44) showed splitting dependencies consistent with our current results, but with less quantitative and specific interpretation in terms of inter- and intra-strand ion exchange and angular analysis. We are not currently aware of (and cannot so far infer) any new exchange rate information pertinent to the double-stranded model. The asymmetry of the satellite peaks at 7% and 8% are attributed to inhomogeneous local alignment of the nematic LC phase. As the average angle approaches 54.7° (magic angle), the change in splitting with angle changes most rapidly (zero crossing, large slope), so any distribution of directors in the nematic phase will cause larger broadening here. Note that nearly all of the spectral peaks for the 7% through 13% spectra are

broader due to this effect. Along with this, any inhomogeneous alignment of LC directors will cause asymmetric quadrupolar splittings. It is conceivable that due to the viscosity of these solutions, not all samples were fully equilibrated in terms of mixing and/or magnetic alignment and that is causing the asymmetry for the 7% and 8% samples. However, lack of perfect equilibration does not generally affect the splitting values in low viscosity aqueous systems such as these, only the linewidths.

The mean square error (0.078) reported for the splittings is the mean-square difference between the fitted value and experimental value. The error bars for $P_2(\cos\theta_Q)$ in the Figure 2f are determined from (and dominated by) the error in the concentration (C) of PBDT polymer in solution, which determines the rod-rod distance ($r \sim C^{-0.5}$) and also $\cos\theta_Q$, where $\cos\theta_Q = P/(P^2 + r^2)^{1/2}$. P is the pitch length and r is the rod-rod distance.

We have modified the text as follows:

Page 10, line 210: Added “In summary, previous double-stranded DNA studies reporting ^{23}Na NMR of counterions^{43,44} showed splitting dependencies consistent with our current results, but with less quantitative and specific interpretation in terms of inter- and intra-strand ion exchange and angular analysis.”

Page 12, line 221 (Figure 2 caption): Added “The error bars for $P_2(\cos\theta_Q)$ are dominated by the error in PBDT polymer weight percent (C) in solution, which determines the rod-rod distance ($r \sim C^{-0.5}$) and also $\cos\theta_Q$, where $\cos\theta_Q = P/(P^2 + r^2)^{1/2}$.”

Comment #4: *This paper would be greatly strengthened by comparison to a control system that does not form the double helix. I realize this could be quite challenging, but might be possible with a different molecular structure (ester or methylated amide), but then the polymer might be different in other ways, such as molecular weight. Is it possible to suppress the double helix formation with a different solvent or with a different counterion? It would also be helpful to confirm that suppressing the double helix reduces the persistence length.*

Response: We very much appreciate these questions and comments. We have studied a very similar control system, and we include additional discussion of this in our revised manuscript. Poly(2,2'-disulfonyl-benzidine isophthalamide) (PBDI) is a non-LC polymer, and the only structural difference from PBDT is the meta linkage in the backbone instead of the para linkage. This relatively minor structural difference causes PBDI aqueous solutions (at any concentration) to exhibit no LC phase and no observed ^{23}Na triplet splittings. This molecule clearly is not a rigid rod and will not form the double helix. Thus, we can demonstrate the *all-para* linkage structure is essential to form the double helix structure. In addition, we have tried many solvents to fully dissolve or denature PBDT (break the double helix), including acetone, toluene, alcohols, THF, ethylene glycol, salt solutions, and DMSO. We found that none of them can dissolve the PBDT into single chains. Additionally, we have changed the counterion of PBDT from Na^+ to Li^+ and the double helix structure is still maintained in all of our measurements (X-ray, NMR, etc...).

We have modified the text as follows:

Page 3, line 64: Added “To compare with the nematic LC phase and discuss the origin of double helices in PBDT, we have studied a chemically similar control system, poly(2,2'-disulfonyl-benzidine isophthalamide) (PBDI). PBDI is a non-LC polymer, and the only structural difference from PBDT is the meta linkage in the backbone instead of the para linkage. This relatively minor structural difference causes PBDI aqueous solutions (at any concentration) to exhibit no NMR, optical or X-ray signatures of a LC phase.³² This molecule is not a rigid rod and will not form the double helix. Thus, we demonstrate that the *all-para* linkage is essential to form the double helix structure.

Page 15, line 294: Added “We note that we have tried many solvents to fully dissolve or denature PBDT and break the double helix, including acetone, toluene, alcohols, THF, ethylene glycol, several salt solutions at varying concentration, and DMSO. However, we found that none of them can dissolve PBDT or break the double helix into single chains.”

Comment #5: MD Simulations

While the results of the simulations look good and the final structures seem to support authors' claims, many details, analysis, and results are missing. In particular, the paper is missing key details about the systems: box size, solvent model and ions added for the first simulations, time steps for both; and the information actually included is spread between the results and the methods sections. While there are some comments in the Results section, the complete details must be added to the Methods section. Many of the methods are missing references (TIP3P water model, AMBER force field, and CM3D software).

Response: We thank the reviewer for these suggestions. Following the advice of the first and second reviewers, we have moved the vacuum phase self-assembly simulations into the **Supplementary Information**. For the solution phase self-assembly simulations, all technical details including time step, box size, and force fields are now provided in the **Methods** section and relevant citations are added in the **References** section. (See also our discussion below in response to Reviewer 3's comments.)

References added to the manuscript:

TIP3P force field water model: (new reference 51)

J. Chem. Phys. **79**, 926-935 (1983).

PACKMOL method for filling simulation box: (new reference 49)

Journal of computational chemistry **30**, 2157-2164 (2009)

Swissparam force field generation for PBDT: (new reference 50)

Journal of computational chemistry **32**, 2359-2368 (2011)

Force fields for sodium ions: (new reference 52)

The Journal of Physical Chemistry B **112**, 9020-9041 (2008)

Ewald summation: (new reference 53)

The Journal of Chemical Physics **98**, 10089-10092 (1993)

Other basic MD simulation methodologies (thermostat, barostat, etc...): (new references 55-58)

Molecular physics **52**, 255-268 (1984)

Physical review A **31**, 1695 (1985).

Journal of Applied physics **52**, 7182-7190 (1981).

Molecular Physics **50**, 1055-1076 (1983).

References added to the *Supplementary Information*:

AMBER force field: (Reference 10 in SI)

J. Comput. Chem. 2004, 25 (9), 1157-1174.

CM3D software: (References 8 and 9 in SI)

J. Chem. Theory Comput. 2007, 3 (3), 1100-1105.

Phys. Rev. Lett. 2010, 105 (23), 237802.

Comment #6: The authors mention on line 249 that they will investigate self-assembly “when two or more chains are present”, but only two are shown. They also mention a set of initial simulations (line 251) but it looks like they just show one. They should discuss whether the results have been replicated in multiple simulations or not, and if so comment on the level of reproducibility.

Response: In the revised manuscript, simulations of self-assembly in vacuum with two and three chains are moved to *Supplementary Information* (Figure S6). The reproducibility of the solution phase self-assembly simulations is explained below in the response to *Comment #9*.

Figure S6. (a) The optimized monomer structure of the PBDT. (b) Crossed hydrogen bonding formed between the amide and sulfonate group. (c, d) The simulation result for two oligomers in 100 ps in vacuum at T = 300 K. (e, f) The simulation result for three oligomers in 100 ps in vacuum at T = 300 K.

Comment #7: How was the optimized structure of PBDT obtained? Was energy minimization done before running any of the simulations? The Methods section should include these details.

Response: In the solution phase self-assembly simulations, after the PBDT molecules were packed with water molecules inside the simulation box, energy minimization was performed using the steepest descent method and the minimization was terminated when the maximal force becomes smaller than $1000 \text{ kJ mol}^{-1} \text{ nm}^{-1}$. This information is provided in the updated *Methods* section.

Comment #8: *For the first simulations, 100 ps looks quite short, but they do not justify why this is sufficient. Also, they comment again about the three chains, but the results are not shown. The SI should include the initial configurations that show up at the beginning and a description of how long the helix configuration lasts.*

Response: We agree with the reviewer that the simulation results in vacuum will not be suitable to include in the main paper. Thus, we have moved all of the first simulation results from the main manuscript to the *Supplementary Information* (Figure S6) as shown in response to Comment #6. For each vacuum simulation at different temperatures, including 10 K, 50 K, 150 K, 250 K, 300 K, 400K, 500 K and 600 K, the runtime was at least 100 picoseconds. The double helix structure will be formed with a simulation temperature above 250 K. In addition, we have allowed enough simulation time up to 300 ps to verify the stability of the formed double helix structure. We discuss the details of the PBDT in water simulations in the following responses.

Comment #9: *Regarding the second method used, I do not understand the sentence: “The estimated pitch length for this model is 3-4 monomer lengths, which is somewhat longer than the first simulation.” Are they trying to justify the difference in length for the systems in the two simulations? They say these simulations are also NVT. Is this correct? In solution, NPT is typically used. Do the different thermodynamic conditions mean the presence of water? If these were actually carried out in NVT ensemble, an explanation for why should be given.*

Response: Yes, the sentence means that the pitch length based on the two simulations are somewhat different. However, based on reviewers’ comments, we decided to move the results of first simulation into *Supplementary Information* and focus on NPT simulations in water for the main manuscript.

The second simulation in the original manuscript was performed in water and in the NVT ensemble. The simulation box is large ($6 \times 10 \times 6 \text{ nm}^3$) and the PBDT polymers occupy a small fraction of the total volume ($< 2\%$). Therefore, the fluctuation of the box size and its impact on the self-assembly should be minor, and using the NVT ensemble should be reasonable.

However, we agree with the reviewer that the NPT simulation is the standard approach for self-assembly in water. Therefore, we repeated the second self-assembly simulations in the NPT ensemble. The two PBDT monomers form a stable double helix structure very similar to that obtained in our original NVT simulations. In the revised manuscript, we only report these NPT simulations and all results described in this response letter are based on these new simulations. The fact that the double helical structure is reproduced in the new simulation lends additional confidence to the reproducibility of our simulations.

Comment #10 – part 1: *Here they say “after energy minimization”, but the method should be specified. Then they say “the system was equilibrated”, but it is not clear if they refer to a simulation previous to the results shown or if they call equilibration the simulation shown here.*

Response: The energy minimization was performed using the steepest descent method until the maximal force in the system was $< 1000.00 \text{ kJ mol}^{-1} \text{ nm}^{-1}$. All simulations performed after the energy minimizations are considered to be equilibration.

We have modified the text as follows:

Page 14, line 267: Deleted “After energy minimization, the system was equilibrated. After ~ 30 ns, the two PBDT molecules intertwined with each other to form a double helical structure as shown in Figure 3d. The system was further equilibrated for ~ 90 ns to ensure the structure is stable.”

Page 14, line 267: Added “After energy minimization, an equilibration simulation was performed. During the first 30 ns, the two PBDT molecules intertwined with each other to form a double helical structure as shown in Figure 3a. The system was equilibrated for an additional ~ 90 ns to ensure the structure is stable.”

Comment #10 – part 2. *Regardless, the pictures in Figure S3 look the same, but a quantitative measure of equilibration must also be shown, such as RMSD, some geometric parameter (like distances between groups that match the experimental data) or SASA, as a function of time. If the system is really equilibrated the plot should reach a plateau. This kind of analysis should also be done for the first system to justify that the simulation time was sufficiently long. I think that another interesting measurement is the extension of the molecules with time.*

These analyses are quite standard, and I am surprised they do not include any, all they did was measuring the distances that matched the experiments but, as I said, this is totally cherry picking, maybe that is the only frame of the simulation where they have that distance. Also, they are quite precise in giving the number 8.4 \AA , but it is not specified how this is obtained (perhaps center of mass to center of mass?).

Response: We thank the reviewer for the helpful suggestions on checking whether the system has reached equilibrium. We performed two analyses to confirm that the double helical structure is stable and equilibrium was reached in our simulations. The below discussion is incorporated into the revised **Supplementary Information**.

First, we calculated the root-mean-square deviation (RMSD) of the double helical structure of PBDTs using the coordinates at ~ 30 ns as a reference structure (the double helical structure was formed by $t = 30 \text{ ns}$). The figure below shows that the RMSD is stable and oscillates around ~ 0.25 nm in a narrow range. The magnitude of the oscillation is similar to prior reports of DNA self-assembly in water (e.g., reference 7 in the new SI – *J. Phys. Chem. B*, 117, 13226, 2013).

Figure S3. The RMSD of the structure formed by the two PBDT monomers in the NPT simulations. The double helical structure was formed by 30 ns, and the structure at this time is taken as the reference structure.

Second, for the self-assembled double helix structure, we compute the distance between the SO_3^- groups at the two ends of the rod-like structure as function of time. This distance shows little drift over a time period of 90 ns (see figure below), thus supporting that the assembled structure has reached equilibrium.

Figure S4. The distance between the SO_3^- groups at the two ends of the self-assembled double helical PBDT structure in the NPT simulation system during the last 90 ns of the 120 ns-long simulation. The double helical structure was formed by 30 ns.

In the revised manuscript, the above two analyses and discussion are incorporated into the *Supplementary Information* (Figure S3 and Figure S4).

In terms of the distance 8.4 \AA between sulfonate groups, in the original manuscript we simply estimated the distance between a few pairs of sulfonate groups (sulfurs) along the helix axis based on the vacuum simulation results, which appeared to agree with the distance from XRD. Note that we have moved all of simulation results in vacuum into *Supplementary Information*. We omitted the mention of this distance from the current paper since there are many sulfonate-sulfonate

distances in the latest NPT calculation and the main point is that any MD simulation conditions employed (water or vacuum, NVT or NPT, different FFs) will self-assemble a double helix.

Comment #11: *SI Figures S3 and S4 show snapshots for the simulation in solution. Figure S3 appears to show the initial and 30 ns data reported in Figure S4. If so, it seems redundant to have both figures, unless Figure S3 is meant to emphasize the organization of water and sodium ions. The main text (line 269 and the legend Figure S4) state “The system was further equilibrated for ~ 90 ns to ensure the structure is stable.” This could be made clearer by saying “for an additional 90 ns...” since the total simulation time labeled in Figure S4 was 120 ns.*

Response: We thank the reviewer for the helpful suggestions. In the revised manuscript, we removed Figure S3 (although the figures have now been redrawn using the data obtained from the NPT simulations in water). We have also reworked Figure 3 in the main paper with relevant molecular structures and snapshots from our revised NPT simulations in order to illustrate the double helix self-assembly. The new Figure 3 and its caption are inserted as follows.

Figure 3. Molecular dynamics (MD) simulations of PBDT polymer chains in water. (a) Simulation result for two oligomers with 4 monomers run for 120 ns in water at $T = 300$ K and $P = 1$ atm. (b-f) Evolution of the conformation of the PBDT monomers during their self-assembly process. The snapshots show the two PBDT polyanions at (b) 0 ns, (c) 10 ns, (d) 20 ns, (e) 30 ns, (f) 120 ns in the simulation. The two PBDT monomers form a double helix structure at ~ 30 ns, and the structure is stable for the rest of the simulation lasting 90 ns. The red and black balls denote the backbone atoms of the two PBDT monomers, and the yellow balls denote the sulfonate group atoms. The blue background denotes the water molecules. Sodium counterions and explicit water atoms are omitted for clarity.

We have modified the text as follows:

Page 14, line 269: Deleted “The system was further equilibrated for ~ 90 ns to ensure the structure is stable.”

Page 14, line 269: Added “The system was equilibrated for an additional ~ 90 ns after the double helix self-assembled to ensure the structure is stable.”

Page 15, line 273: Deleted *original Figure 3 caption* “**Figure 3.** Molecular dynamics (MD) simulations of PBDT polymer chains. (a) Optimized monomer structure of PBDT. (b) Simulation result for two oligomers run for 100 ps in vacuum at T = 300 K. (c) Self-assembled double helix with a pitch approximately equal to the XRD results and model of Figure 1. (d) Further MD simulation result in water using a different simulation method, which exhibits a double helix with a somewhat longer helical pitch length.”

Page 17, line 336: Deleted: *Original MD simulations Methods section* “In the first MD study, simulations of the polymer chains in vacuum were all accomplished with the CM3D molecular dynamics program. The force field was the Assisted Model Building with Energy Refinement (AMBER), which is often used for simulating proteins and DNA. Periodic boundary conditions (PBC) were used and a Nosé-Hoover thermostat at 300 K was used for simulations at a constant temperature.

The second MD simulation study was performed to investigate the self-assembly of two initially separated PBDT oligomers in aqueous solutions. Na⁺ counterions were included in the system to ensure electroneutrality. The simulation box was periodic in all three directions and had a size of 6×10×6 nm³. The force field parameters for the PBDT polyanions were generated using the Swissparam server.⁴³ The TIP3P model was employed for the water molecules. The force fields for the sodium ions were taken from the OPLS-AA force field.⁴⁴ The system temperature was maintained at 300 K using the Nosé-Hoover thermostat. The non-electrostatic interactions were computed via direct summation with a cut-off length of 1.2 nm. The electrostatic interactions were computed using the Particle Mesh Ewald (PME) method with a real-space cutoff length of 1.2 nm. All bonds were constrained using the LINCS algorithm.⁴⁵”

Page 17, line 336: Added *Revised MD simulations Methods section*

“MD simulations were performed to investigate the self-assembly of two initially separated PBDT oligomers in aqueous solution. Two PBDT monomers, each with 4 repeating units and measuring ~ 6.8 nm in length, were initially placed side-by-side in a 6×6×10 nm³ simulation box. The simulation box was filled with 11850 water molecules using the Packmol code.⁴⁹ 16 Na⁺ ions were included to ensure the electro-neutrality of the system. The simulation box was periodic in all three directions. The force field parameters for the PBDT polyanions were generated using the Swissparam server.⁵⁰ The TIP3P model was employed for the water molecules.⁵¹ The force fields for the sodium ions were taken from the work by Joung and Cheatham.⁵²

Simulations were performed using an MD code Gromacs 4.5.⁴⁸ First, an energy minimization was conducted using the steepest descent method, and the minimization was terminated when the maximal force in the system became smaller than 1000.00 kJ mol⁻¹ nm⁻¹. Next, a 120 ns equilibrium simulation was performed in the NPT ensemble using a time step size of 2 fs. The non-electrostatic interactions were computed via direct summation with a cut-off length of 1.2 nm. The electrostatic interactions were computed using the Particle Mesh Ewald (PME) method.⁵³ The

real space cutoff and FFT spacing were 1.2 nm and 0.12 nm, respectively. All bonds were constrained using the LINCS algorithm.⁵⁴ The system temperature was maintained at 300 K using the Nose-Hoover thermostat (time constant: 1 ps)^{55,56} and the pressure was maintained at 1atm using the Parrinello-Rahman barostat (time constant: 10 ps).^{57,58} The trajectory was saved every 10 ps and analyzed using the tools provided by Gromacs.”

Response to Reviewer 3's Comments

Statement: *This is an interesting paper, reporting a double helix macromolecule with a rigid and simple molecular structure. As I am a computational biophysicist, I address here only general points of the paper as well as the MD simulations.*

Response: We thank the reviewer for this succinct summary of our work, and we have revised the paper based on these helpful comments.

Comment #1: *Some sentences are redundant and obvious. For instance: X-ray diffraction (XRD) is commonly used to determine the crystalline or semi-crystalline structure of small molecules, proteins and macromolecules“. Please remove them.*

Response: Thanks for this comment. We have removed this sentence from our manuscript.

We have modified the text as follows:

Page 3, line 71: Deleted “X-ray diffraction (XRD) is commonly used to determine the crystalline or semi-crystalline structure of small molecules, proteins and macromolecules.”

Comment #2: *Some sentences can be shortened without any loss of information. For instance, replace “In 1953, the DNA double helical configuration was first proposed by Watson and Crick^{1,2} based on an XRD pattern of DNA fibers by Rosalind Franklin.²⁹ Following a similar approach, we also employ XRD to study the packing structure and morphology of PBDT aqueous solutions.“ with „Following Watson and Crick’s approach for the DNA double helix (1,2), we employ XRD to study the packing structure and morphology of PBDT aqueous solutions” .*

Response: Thanks for this suggestion. However, we think the sentence we use will introduce more background information about the DNA double helix structure. We prefer to retain our original statement.

Comment #3: *Can the authors write a modified version of the HELIX program so that it can differentiate the –SO₃– and –NHCO– functionalities? This would address an important point of the paper.*

Response: We thank reviewer for the significant questions and comments. We agree that it would be exciting if HELIX could differentiate –SO₃– and –NHCO– functionalities. However, we do not currently have the expertise or resources to write a reliable modified version of this program. The HELIX package was developed by Carlo Knupp and John M. Squire. In future work, we will endeavor to match exactly the XRD data with a simulation, and we are beginning to do this as part of another collaboration. However, we feel that the data is conclusive at this point, warranting the conclusion that PBDT is a double helix.

Comment #4: *As for the MD simulations, the authors should report only the results for the simulation in water. This referee is highly skeptical that the authors can obtain any reliable information for their anionic polymer by performing simulations in vacuo. This has been amply demonstrated for DNA double helix simulations.*

Response: We thank the reviewer for raising this question, which was also pointed out by the other reviewers. In the revised manuscript, we have moved the preliminary vacuum simulations into *Supplementary Information*.

Comment #5: *In the MD simulations in water:*

- *What is the pressure?*
- *Why didn't the authors perform simulation in the canonical ensemble?*
- *Can the authors describe the counterions distribution quantitatively ?*
- *What is the rationale based on which they used the OPLS force field? Has this force field already been tested for systems similar to that used here?*
- *What was the time step used?*
- *How did the authors assess quantitatively that the structure was equilibrated (see SI, "In Figure S3, after energy minimization, the system was equilibrated")?*

Response: We thank the reviewer for examining our simulations in depth. Below we respond to each question separately.

- *What is the pressure?*
- *Why didn't the authors perform simulation in the canonical ensemble?*

The self-assembly simulation in water in the original manuscript was performed in the NVT ensemble. Because the simulation box is large ($6 \times 10 \times 6 \text{ nm}^3$) and the PBDT polymers occupy a small fraction of the total volume ($< 2\%$), the fluctuation of the box size and its impact on the self-assembly should be minor. Therefore, we used the NVT ensemble. Nevertheless, we agree with the reviewer that the NPT simulation is the standard approach for self-assembly in water. Therefore, we repeated the second self-assembly simulations in the NPT ensembles ($P = 1 \text{ atm}$, $T = 300 \text{ K}$). The two PBDT monomers form a stable double helix structure very similar to that obtained in original our NVT simulations. In the revised manuscript, we only report these NPT simulations and all results described in this response letter are based on these new simulations. The fact that the double helical structure is reproduced in the new simulations lends confidence to the reproducibility of our simulations and the robustness of this double helical structure for PBDT.

- *Can the authors describe the counterions distribution quantitatively ?*

We computed the distribution of the Na^+ counterions around the sulfonate groups in the PBDT. The results are shown in Figure S5a. As a reference, the distribution of water molecules around the sulfonate groups is also shown in Figure S5b. We have included this Figure S5 in the revised *Supplementary Information*.

Figure S5. The density distribution of Na^+ ions (a) and water molecules (b) around the sulfur atoms of PBDT's sulfonate groups. These densities are based on the last 90 ns of the NPT simulation. The position of water molecules is based on their oxygen atom.

We have added the following text:

Page 14, line 270: Added “Additionally, we computed the distribution of the Na^+ counterions around the sulfonate groups in the PBDT, and this result is shown in Figure S5a. As a reference, the distribution of water molecules around the sulfonate group is also shown in Figure S5b.”

-What is the rationale based on which they used the OPLS force field? Has this force field already been tested for systems similar to that used here?

We used the OPLS force field for the Na^+ ion because a systematic study of ion force fields suggests that this force field produces reasonable results for ion hydration (*J. Comput. Chem.* 25, 678, 2004.). However, we recognize that this force field is not the standard choice when the TIP3P water model is adopted. Therefore, in our new NPT simulation, we adopted the widely used Na^+ force fields developed by Joung and Cheatham (*J. Phys. Chem. B*, 112, 9020, 2008). The new simulation produced double helical structure very similar to that found in our previous simulations. Indeed, for any conditions we have tried (at least four different force fields, as well as in water and in vacuum), PBDT self assembles into a double helix.

-What was the time step used?

The time step is 2 fs. This is now stated in the revised **Methods** section.

-How did the authors assess quantitatively that the structure was equilibrated (see SI, “In Figure S3, after energy minimization, the system was equilibrated”)?

In the revised manuscript, we performed two sets of analyses to quantitatively assess whether the structure was indeed equilibrated. Please refer also to our *Response to Reviewer 2, Comment #10*.

In the revised manuscript, the above response has been incorporated into the **Methods** section and in the **Supplementary Information** (Figures S3 and S4).

Reviewers' comments:

Reviewer #1 (Remarks to the Author):

The authors have made their best efforts to answer the points I raised as well as those raised by the other two reviewers and have modified their manuscript accordingly. I think both the answers and modifications are satisfactory and I support publication at this stage. The authors might want to consider the comments below concerning references:

Refs 8 & 24 have been updated recently: Chem. Rev. 2016, 116, 13752–13990. This latter review contains a comprehensive section on double helices. The authors might want to go through it and eventually complement the citations on double helices in their introduction.

Reference 14 is wrongly called as being relevant to peptide nucleic acids while it concerns beta-peptides. From the same authors, an important reference on double helices has been overlooked: Angew.Chem. Int.Ed. 2018, 57,1057–1061

Reviewer #2 (Remarks to the Author):

I am generally satisfied with the changes made in the manuscript based on the reviewers' comments and I recommend publication after minor revisions. I have few additional comments regarding the revised manuscript:

- 1) The reference to PBDI as a non-LC polymer is a helpful addition to the paper.
- 2) On page 9, line 174, the authors refer to B and C of Figure 1b. Should this reference to be for Figure 1a. If they do indeed mean Figure 1b, it would be better to indicate B and C in that panel.
- 3) The authors should specify how the error bars in Figure 2f were calculated.
- 4) The new Figure 3 looks very nice, but the resolution in panels b through f is poor and should be improved.
- 5) The stability of the double helix is impressive and the distance between groups which are relatively far apart supports this stability. In my opinion, this is a sufficiently interesting result to add it to the main text (after improving the quality of the plot a bit and adding a schematic to show the distance it refers to). In the last paragraph before the conclusions, the authors offer a hypothesis for all these interactions. I recommend that they provide some quantitative analysis of nature of these interacting groups. Figure S6b shows the presence of good H-bonding geometries, so I am confident that they could get very good quantitative data (easily carried out in GROMACS). This would reinforce the claimed stability of the helix and would make the simulation part of the paper more rigorous.
- 6) Figure S6f: I do not see the third oligomer, which based on Figure S6e should appear in red.
- 7) In general, the Simulations part of the Supplementary Information could be more complete. The paper would benefit from extra snapshots and zoomed images, as in Figure S6b. The zoomed images can be of interest to better understand the geometry of the helix and could help refine the experimental models. The extra snapshots would support help support the conclusions of the paper. I understand that you do not fill the manuscript with useless and highly repetitive images, but I would expect there to be more detailed and systematic examples (e.g., putting snapshots every 10 ns) in the SI.
- 8) Following the previous comment, I would have liked to see snapshots of the results used to calculate the RDFs, including the Na⁺ ions. Some of these snapshots were in the prior to the revisions. Also, the studies of the assemblies at different temperatures are of methodological interest. These results would further support the quality of the work.

Reviewer #3 (Remarks to the Author):

While I highly appreciate the effort of the authors to most of my points, it was disappointing that

the authors could not properly address one of the really important issues: "Can the authors write a modified version of the HELIX program so that it can differentiate the $-SO_3^-$ and $NHCO_2^-$ functionalities?" The authors themselves have recognized the importance of my remark in their reply: "We thank reviewer for the significant questions and comments."

Without this, I think that the paper, still very valid, lacks a key piece of information, which is required for the very diverse and general readership of the Journal. The paper at the present stage is more suitable for a more specialized journal than Nature Communication.

Minor point:

My issue "Some sentences can be shortened without any loss of information" has not been addressed.

Response to Reviewer 1's Comments

Statement: *The authors have made their best efforts to answer the points I raised as well as those raised by the other two reviewers and have modified their manuscript accordingly. I think both the answers and modifications are satisfactory and I support publication at this stage.*

Response: We appreciate the reviewer's support for publication of this work in Nature Communications.

Comment #1: *The authors might want to consider the comments below concerning references: Refs 8 & 24 have been updated recently: Chem. Rev. 2016, 116, 13752–13990. This latter review contains a comprehensive section on double helices. The authors might want to go through it and eventually complement the citations on double helices in their introduction. Reference 14 is wrongly called as being relevant to peptide nucleic acids while it concerns beta-peptides. From the same authors, an important reference on double helices has been overlooked: Angew.Chem. Int.Ed. 2018, 57,1057–1061*

Response: We thank the reviewer for this suggestion and for pointing out these previous related works. We have gone through these references and added two new references to the manuscript. We have also corrected the citation of reference 14 by adding a new category in the sentence describing the structural platforms.

We have modified the text as follows:

Page 2, line 39:

Original:

“The most commonly encountered structural motifs in oligomers that enable double helical formation are based on peptide nucleic acids,^{13,14} amidinium-carboxylate salt bridges,^{6,8} coordination polymers.^{7,9,15-17”}

Modified:

“The most commonly encountered structural motifs in oligomers that enable double helical formation are based on **peptides**,^{14,15} peptide nucleic acids,¹⁶ amidinium-carboxylate salt bridges,^{6,8} and coordination polymers.^{7,9,17-19”}

References added to the manuscript: (number corresponds to the current revised manuscript)

13. Yashima, E. *et al.* Supramolecular Helical Systems: Helical Assemblies of Small Molecules, Foldamers, and Polymers with Chiral Amplification and Their Functions. *Chemical Reviews* **116**, 13752-13990.

14. Misra, R., Dey, S., Reja, R. M. & Gopi, H. N. Artificial β -Double Helices from Achiral γ -Peptides. *Angewandte Chemie International Edition* **57**, 1057-1061, doi:doi:10.1002/anie.201711124 (2018).

Response to Reviewer 2's Comments

Statement: *I am generally satisfied with the changes made in the manuscript based on the reviewers' comments and I recommend publication after minor revisions. I have few additional comments regarding the revised manuscript.*

Response: We thank the reviewer for supporting publication of our work. Below please find our responses to these new suggestions and comments.

Comment #1: *The reference to PBDI as a non-LC polymer is a helpful addition to the paper.*

Response: We thank reviewer for this comment, and agree this significantly strengthens the paper.

Comment #2: *On page 9, line 174, the authors refer to B and C of Figure 1b. Should this reference to be for Figure 1a. If they do indeed mean Figure 1b, it would be better to indicate B and C in that panel.*

Response: We thank reviewer for pointing out this error. It should indeed be Figure 1a instead of Figure 1b, and we have modified the text accordingly (**Page 9, line 174**).

Comment #3: *The authors should specify how the error bars in Figure 2f were calculated.*

Response: We thank reviewer for this question. We had briefly introduced quantification of the error bar in the caption of Figure 2, but we agree this should be made more clear.

We have expanded the text as follows:

Page 3, line 239 (Figure 2 caption):

Original:

“The error bars for $P_2(\cos\theta_Q)$ are dominated by the error in PBDT polymer weight percent (C) in solution, which determines the rod-rod distance ($r \sim C^{-0.5}$) and also $\cos\theta_Q$, where $\cos\theta_Q = P/(P^2 + r^2)^{1/2}$.”

Modified:

“The error bars for the $P_2(\cos\theta_Q)$ model points shown are dominated by the standard deviation in PBDT polymer weight percent (C) in solution (*ca.* 2.5%) as a percentage of the total concentration of polymer in each solution (*e.g.*, 0.025×5 wt% polymer). C determines the rod-rod distance ($r \sim C^{-0.5}$) and also $\cos\theta_Q$, where $\cos\theta_Q = P/(P^2 + r^2)^{1/2}$. Thus, the error bars are calculated based only on the error in PBDT concentration.”

Comment #4: *The new Figure 3 looks very nice, but the resolution in panels b through f is poor and should be improved.*

Response: We thank reviewer for this comment. We have attempted to guarantee high resolution for Figure 3 parts b-f based on the journal's requirements. The resolution of the figures may have been decreased in the PDF version of the document sent out for review, since when we look at our Word version as submitted it exhibits much higher resolution than the “proof” PDF. We will work with the editorial staff to make sure the final figures have sufficiently high resolution.

Comment #5: *The stability of the double helix is impressive and the distance between groups which are relatively far apart supports this stability. In my opinion, this is a sufficiently interesting*

result to add it to the main text (after improving the quality of the plot a bit and adding a schematic to show the distance it refers to). In the last paragraph before the conclusions, the authors offer a hypothesis for all these interactions. I recommend that they provide some quantitative analysis of nature of these interacting groups. Figure S6b shows the presence of good H-bonding geometries, so I am confident that they could get very good quantitative data (easily carried out in GROMACS). This would reinforce the claimed stability of the helix and would make the simulation part of the paper more rigorous.

Response: We thank the reviewer for these suggestions. We agree that the simulation *done in vacuum* as shown in Figure S6 (in the original submission) is interesting. However, the core concept in this paper involves investigating PBDT polymer in water, and these simulations (Figure 3 in main document, and Figures S3, S4, and S5 in SI), form the main basis of our relevant results. According to the first round of reviews, all reviewers heavily criticized the significance of the simulation done in vacuum (Figure S6). We decided to keep this preliminary simulation in Supplementary Information instead of removing it entirely from the paper as suggested by one reviewer. At this point, after discussion among our co-authors and based on the suggestions of other reviewers, we think it is still suitable to keep the vacuum simulation results in Supplementary Information and yet view them with extreme caution regarding the detailed implications of their structural parameters. Therefore, we do not feel it is prudent to do further structural analysis on this vacuum simulation since it does not correspond to any experimental results. Regarding quantitative analysis of the detailed structure and interactions of PBDT in water, we are continuing in future investigations to use additional structural techniques, including solid-state NMR, X-ray diffraction, cryo-TEM, and simulations. We have also added new simulation figures in our revised SI (See Comment #7 response below) to further augment interpretation of our results.

Comment #6: *Figure S6f: I do not see the third oligomer, which based on Figure S6e should appear in red.*

Response: We thank reviewer for this question about the vacuum simulations. During the simulation with three oligomers, two of the three form a double helix at 100 ps. Meanwhile, the third (red) chain has no interaction with the other two and slowly moves away, though it is initially within the reach of the Van der Waals and electrostatic interactions. We have added Figure S10 in Supplementary Information, which shows the double helix with the third (decoupled) oligomer in the background.

Figure S10. The simulation results with three chains in the system (Figure S9e). Two chains form a double helix with the third chain in the background.

Comment #7: *In general, the Simulations part of the Supplementary Information could be more complete. The paper would benefit from extra snapshots and zoomed images, as in Figure S6b. The zoomed images can be of interest to better understand the geometry of the helix and could help refine the experimental models. The extra snapshots would support help support the conclusions of the paper. I understand that you do not fill the manuscript with useless and highly repetitive images, but I would expect there to be more detailed and systematic examples (e.g., putting snapshots every 10 ns) in the SI.*

Response: We thank reviewer for these suggestions. We have added zoomed views of the simulations with Na⁺ counterion in water (Figure S7) and more snapshots for every 10 ns after double helix formation (Figure S8) to the Supplementary Information. We have also inserted snapshots of the initial assembly event in water similar to those that appeared in the originally submitted manuscript (but with added Na⁺ counterions), as the new Figure S6.

Figure S7. Snapshots at 30 ns and 70 ns of two representative frames (top and side view) used when calculating the radial density of Na⁺ ions around the sulfur atoms of the PBDT sulfonate groups shown in Figure S5. Blue balls denote Na⁺ ions, which in these snapshots are located at approximately the most probable radial distance from the -SO₃⁻ anions (first peak in Figure S5).

Figure S8. Simulation results for two oligomers with 4 monomers in water at $T = 300$ K with simulation times of 40 ns, 50 ns, 60 ns, 70 ns, 80 ns, 90 ns, 100 ns, and 110ns.

Figure S6. Self-assembly of PBBDT monomers into a double helix structure. (a) Two PBBDT monomers are initially packed side-to-side in water with Na⁺ ions. (b) After ≈ 30 ns, the two monomers intertwine with each other to form a double helix structure. The red and black balls denote the backbone of the two PBBDT monomers and the yellow balls denote the sulfonate groups. The cyan dots and the blue balls denote the water molecules and the Na⁺ ions, respectively. Only a portion of the simulation box and the water/ions in the box are shown for clarity.

Comment #8: *Following the previous comment, I would have liked to see snapshots of the results used to calculate the RDFs, including the Na⁺ ions. Some of these snapshots were in the prior to the revisions. Also, the studies of the assemblies at different temperatures are of methodological interest. These results would further support the quality of the work.*

Response: We thank the reviewer for these suggestions. We have added more snapshots associated with PBBDT chains and Na⁺ ion distributions as shown in Figure S6, S7, and S8 and as mentioned in the response to Comment #7 above. Regarding the simulation at different temperatures, we will continue in future investigations to understand this double helical system by looking at additional features of simulations, including dependencies on temperature, salt content and pH of the PBBDT aqueous solution.

Response to Reviewer 3's Comments

Statement: *While I highly appreciate the effort of the authors to most of my points, it was disappointing that the authors could not properly address one of the really important issues.*

Without this, I think that the paper, still very valid, lacks a key piece of information, which is required for the very diverse and general readership of the Journal. The paper at the present stage is more suitable for a more specialized journal than Nature Communication.

Response: We thank the reviewer for this succinct summary of our work.

Comment #1: *"Can the authors write a modified version of the HELIX program so that it can differentiate the $-SO_3^-$ and $NHCO_2^-$ functionalities?" The authors themselves have recognized the importance of my remark in their reply: "We thank reviewer for the significant questions and comments."*

Response:

We thank the reviewer for bringing up this question again. Currently we do not have the expertise (programming background) or resources (raw code to build up new software) to write and sufficiently vet a modified version of this program. We agree that this would improve the details involved in this study, and as mentioned previously we are beginning to explore new collaborations to develop programs to match exactly the XRD data with a simulation that includes all molecular and orientational features. However, we expect it will take significant time to get the right simulation answer for the right reasons.

We would like to re-emphasize (see *page 9, line 172* in the original document) that there is partial disorder (orientational order parameter ≈ 0.78) along the chain axes due to the nematic liquid crystalline nature of the PBDT solutions. We believe it is this feature that produces the largest deviations between the HELIX simulation (which assumes orientational order parameter = 1) and our data. Liquid crystals such as PBDT solutions present unique problems for interpreting and simulating XRD data that are not observed in perfectly crystalline (or isotropic liquid) systems. There are only a few groups in the world with specific expertise in simulating liquid crystalline aspects of XRD patterns and so we do not take on such an extension to this project lightly.

Perhaps the reviewer might consider that we are presenting novel NMR and X-ray diffraction measurements, along with atomistic simulations involving multiple force fields and conditions (water, vacuum, NVT, NPT). Each of these disparate techniques provides substantial and corroborating evidence for the double helical structure. We are attempting to maintain a high level of understanding of every experimental and computational aspect of this study. Adding an additional programming component to the present study, while certainly valid and desirable, introduces another dimension in terms of vetting programs and results, and most likely a new set of collaborators.

Comment #2: *Minor point:*

My issue "Some sentences can be shortened without any loss of information" has not been addressed.

Response: We thank reviewer for this comments. We have carefully worked through the manuscript and shortened and clarified some wordy sentences.

We have modified the text as follows:

Page 2, line 29:

Original:

“DNA molecules, which act as a storage and transfer platform for the genetic information of life, exhibit a double-stranded helical conformation that has been known for decades.^{1,2}”

Modified:

“DNA molecules, which act as a storage and transfer platform for the genetic information of life, exhibit a double-stranded helical conformation.^{1,2}”

Page 4, line 82:

Original:

“In 1953, the DNA double helical configuration was first proposed by Watson and Crick^{1,2} based on an XRD pattern of DNA fibers by Rosalind Franklin.³⁵ Following a similar approach, we also employ XRD to study the packing structure and morphology of PBDT aqueous solutions.”

Modified:

“The DNA double helical configuration was proposed by Watson and Crick^{1,2} based on an XRD pattern of DNA fibers by Rosalind Franklin.³⁵ Here, we employ XRD to study the packing structure and morphology of PBDT aqueous solutions.”

Page 4, line 94:

Original:

“Indeed, based on the concentration above which the aligned phase forms (1.5 wt%)³⁴ the persistence length (stiffness) along the PBDT rod axis³⁶ is > 240 nm based on Onsager theory,³⁷ and ~ 670 nm based on Flory theory³⁸ (see also Supplementary Information). ”

Modified:

“Indeed, the aligned phase forms above a critical concentration (1.5 wt%),³⁴ where the persistence length (stiffness) along the PBDT rod axis³⁶ is > 240 nm (Onsager theory),³⁷ and ~ 670 nm (Flory theory).³⁸ See SI for additional details.”

Page 5, line 99:

Original:

“With refinements in the synthesis of PBDT beyond our original study, yielding a higher molecular weight (see SI for details), we have observed that the aligned phase can form at concentrations down to 0.3 wt% PBDT in water, which represents a persistence length of > 1.2 μm . ”

Modified:

“With refinements in synthesis, we obtain higher molecular weight PBDT (see SI for details). The aligned phase forms at concentrations down to 0.3 wt% PBDT, which represents a persistence length of > 1.2 μm .”

Page 11, line 223:

Original:

“In summary, previous double-stranded DNA studies reporting ²³Na NMR of counterions^{44,45} showed splitting dependencies consistent with our current results, but with less quantitative and specific interpretation in terms of inter- and intra-strand ion exchange and angular analysis.”

Modified:

“In summary, compared to previous double-stranded DNA studies, the ^{23}Na NMR of counterions^{44,45} here shows splitting dependencies that provide more quantitative interpretation in terms of inter- and intra-strand ion exchange.”

REVIEWERS' COMMENTS:

Reviewer #2 (Remarks to the Author):

I am satisfied by the authors' responses to all three reviewers.

Of particular note, Reviewer 3 says: "

While I highly appreciate the effort of the authors to most of my points, it was disappointing that the authors could not properly address one of the really important issues: "Can the authors write a modified version of the HELIX program so that it can differentiate the –SO₃ - and NHCO– functionalities?"

I think that this was a good suggestion and the authors recognized that as well. However, the authors do not have the resources to fully address this concern and I do not think it is necessary for this paper to be published.

Reviewer #3 (Remarks to the Author):

I agree with the authors that writing a modified version of the HELIX program so that it can differentiate the –SO₃ - and NHCO– functionalities is a significant question and it would improve the details involved in this study. Unfortunately, this important issue has not been addressed in the revised version of the manuscript. Hence, as I wrote in my previous review, I think that the paper in its present form is more suitable for a specialized journal rather than Nature Communications.